# Structural delineation and computational design of SARS-CoV-2-neutralizing antibodies against Omicron subvariants

Saya Moriyama [1] ✉, Yuki Anraku [2], Shunta Taminishi[3], Yu Adachi[1], Daisuke Kuroda [1], Shunsuke Kita [2], Yusuke Higuchi[3], Yuhei Kirita [4], Ryutaro Kotaki [1], Keisuke Tonouchi[1,5], Kohei Yumoto[1], Tateki Suzuki[6], Taiyou Someya[2], Hideo Fukuhara [7], Yudai Kuroda[8], Tsukasa Yamamoto[8], Taishi Onodera[1], Shuetsu Fukushi[9], Ken Maeda [8], Fukumi Nakamura-Uchiyama[10], Takao Hashiguchi [6], Atsushi Hoshino [3], Katsumi Maenaka [2,7,11,12] & Yoshimasa Takahashi [1] ✉

SARS-CoV-2 Omicron subvariants have evolved to evade receptor-binding site (RBS) antibodies that exist in diverse individuals as public antibody clones. We rationally selected RBS antibodies resilient to mutations in emerging Omicron subvariants. Y489 was identified as a site of virus vulnerability and a common footprint of broadly neutralizing antibodies against the subvariants. Multiple Y489-binding antibodies were encoded by public clonotypes and additionally recognized F486, potentially accounting for the emergence of Omicron subvariants harboring the F486V mutation. However, a subclass of antibodies broadly neutralized BA.4/BA.5 variants via hydrophobic binding sites of rare clonotypes along with high mutation-resilience under escape mutation screening. A computationally designed antibody based on one of the Y489-binding antibodies, NIV-10/FD03, was able to bind XBB with any 486 mutation and neutralized XBB.1.5. The structural basis for the mutation-resilience of this Y489-binding antibody group may provide important insights into the design of therapeutics resistant to viral escape.

Since the emergence of severe acute respiratory syndrome coronavirus 2 (SARS-CoV-2) in 2019, the coronavirus disease 2019 (COVID-19) pandemic has been a global health issue. SARS-CoV-2 binds the host entry receptor angiotensin-converting enzyme 2 (ACE2) with the spike glycoprotein. Neutralizing antibodies against SARS-CoV-2 spike glycoprotein, primarily the receptor-binding domain (RBD), are essential for COVID-19 prevention and treatment[1–3]. Mutations in the RBD of spike proteins negatively affect the potency of neutralizing antibodies in convalescent/vaccinated plasma and therapeutic monoclonal antibodies, thereby contributing to increased infectivity and spread[4].

In November 2021, the Omicron BA.1 variant, with more than 15 RBD mutations, was identified and detected globally. Drastic antigenic changes in this variant hindered the neutralizing activity of many therapeutic antibodies used clinically[5,6]. Thereafter, the BA.1 variant was out-competed by the Omicron BA.2 variant, which was characterized by increased transmissibility and antibody evasion. BA.2 was the dominant variant worldwide in spring 2022, but BA.2-related variants (BA.2.12.1, BA.4, BA.5, and BA.2.75) emerged with additional RBD mutations: L452Q/R in BA.2.12.1, BA.4, and BA.5; F486V in BA.4 and BA.5; G446S/N460K in BA.2.75; and R493Q in BA.4, BA.5 and BA.2.75. The BA.2 variant was out-competed by the BA.5 variant in many

countries, including Japan. New mutations in BA.4/5 further exaggerated the ability of the virus to escape polyclonal neutralizing antibodies afforded by previous vaccination and infection[7–9]; this phenomenon implies viral evolution under immune pressure by herd (public) immunity. Therefore, it is crucial to clarify the relationship between emerging RBD mutations and the footprints of Omicron-neutralizing antibodies that exist in diverse individuals as public clonotypes. This information is required to prepare neutralizing antibodies that are effective for emerging Omicron variants and those that may arise in the future.

Here, we employed a monoclonal antibody screening method that recognizes sites of virus vulnerability, so that the identified antibodies resist emerging variants, including Omicron subvariants BA.4/BA.5. As the vulnerable sites, we focused on ACE2-binding sites in the receptor-binding domain (namely receptor-binding site, RBS), which have pivotal functions in viral entry into host cells. Furthermore, sites in the RBS that are scarcely mutated in circulating viruses in the real world are more likely to be virus-vulnerable sites. Indeed, several RBS antibodies with broad neutralizing activity have been found to recognize such vulnerable sites[10,11]. Using recombinant RBDs with single amino acid substitutions, we successfully identified virus-vulnerable site tyrosine-489 (Y489)-recognizing monoclonal antibodies from COVID-19 convalescent PBMCs. Y489-targeting antibodies can be subdivided by their dependence on phenylalanine-486 (F486) binding and Y489-targeting, but F486-non-targeting (Y489+F486−) antibodies showed neutralization against multiple recent variants, including BA.4/5 and BA.2.75. Y489+F486+ antibodies could neutralize the ancestral strain, Delta variant, BA.1, BA.2, and BA.2.75, but were susceptible to BA.4/5. One of the Y489+F486− antibodies, NIV-10, had a high resistance to escape by single amino acid substitution in the RBD. Structural data revealed the relatively broad epitope of the NIV-10 antibody. NIV-10 and other Y489+F486− antibodies had a hydrophobic heavy chain complementarity-determining region 3 (H-CDR3) compared to the

Y489+F486+ clones. H-CDR3 may facilitate the interaction with the hydrophobic site in the RBD with a broad epitope, leading to high escape resistance. NIV-10 potently neutralized one of the recent Omicron subvariants, BQ.1.1. However, the P486 mutation impeded NIV-10 binding to XBB.1.5. To improve the attenuated neutralizing activity of NIV-10 against XBB sublineages, we utilized a computational design and successfully generated an engineered NIV-10-derivative antibody, NIV-10/FD03, which could bind to RBDs with various mutations, including XBB.1.5 with the neutralizing activity. Our study highlights Y489+F486− antibodies as a target for broadly neutralizing boosting vaccines and as reasonable templates for designing new and effective antibody therapeutics.

## Results

### Tyrosine-489 in the RBD is a key epitope of variant-neutralizing antibodies

We rationally screened neutralizing antibodies against ACE2-binding sites poorly mutated in the real world. ACE2-binding sites are colored based on the mutation frequency among the circulating virus sequences deposited in the GISAID database (Fig. 1a). Among low-frequency mutations within or near RBS, we selected seven amino acid mutations, S443N, G447H, F456D, N487G, Y489S, T500D, and G502M, as RBD mutants for the initial antibody screening (Fig. 1a, b). These mutations have been reported to attenuate the ACE2-binding ability[12]. Indeed, seven recombinant RBDs with these mutations had undetectable binding affinities to human ACE2 (Supplementary Fig. 1a).

RBD-binding monoclonal IgG antibodies ($n = 947$) were prepared from memory B cells in peripheral blood mononuclear cells (PBMCs) of COVID-19 convalescent individuals (Supplementary Fig. 1b and Supplementary Table 1). Blood was drawn at >7 months after infection to allow sufficient time for antibody evolution[13], and one participant received two mRNA vaccinations before the blood draw. To enrich mutation-resistant B cells, the Beta-variant RBD probe with the largest

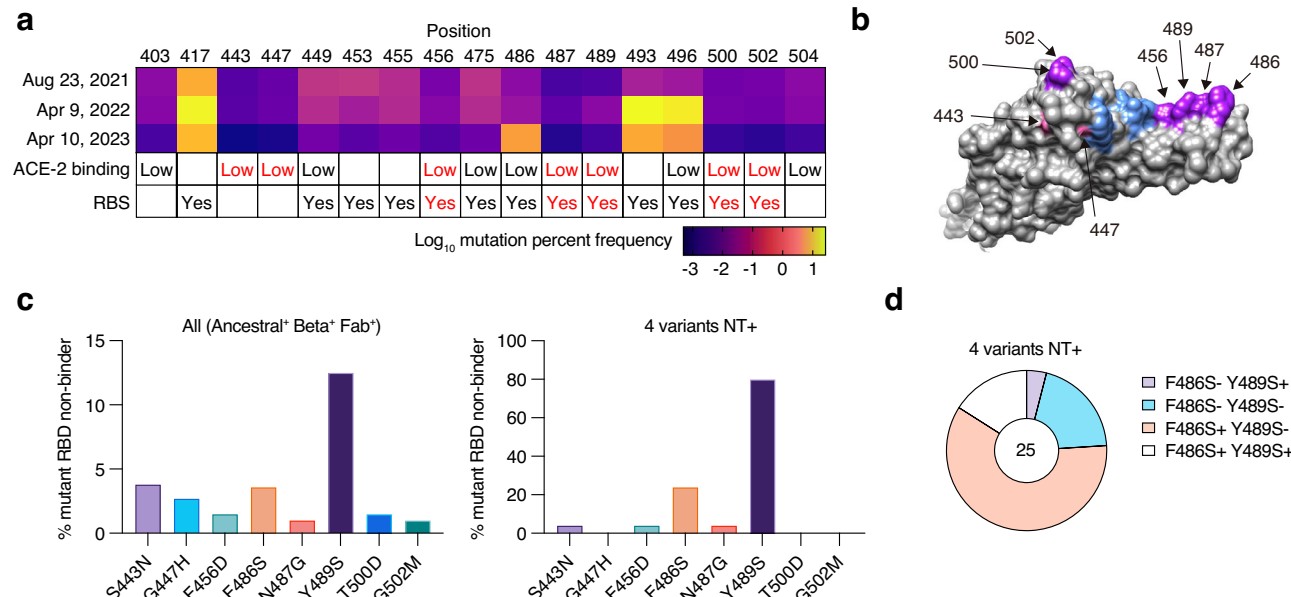

**Fig. 1 | Monoclonal antibody screening with RBD harboring a point mutation from convalescent PBMCs. a** Mutation frequency calculated based on GISAID record from August 2021 to April 2023, position of the amino acid with low ACE2 binding[12], and receptor-binding site (RBS) is shown. Mutations at seven positions highlighted in red were used for further analysis. **b** The position of the seven mutations and RBS are presented in the RBD graphics. Five out of seven mutations and 486 in RBS are highlighted in purple, two out of seven mutations outside RBS are highlighted in pink, and other RBS are presented in blue. **c** Percentages of

clones with binding to indicated mutant RBD below the threshold (non-binder) among ancestral RBD and Beta RBD double-binding clones are shown on the left (823 double-binding clones in total), and percentages of clones with binding under the threshold among clones neutralizing (NT+) all ancestral, Beta, Delta, and Omicron variants are shown on the right (25 clones in total). **d** Pie chart of single B-cell culture clones with F486S non-binding (F486S−), F486S binding (F486S+), Y489S non-binding (Y489S−), or Y489S binding (Y489S+) among neutralizing clones for four variants is shown.

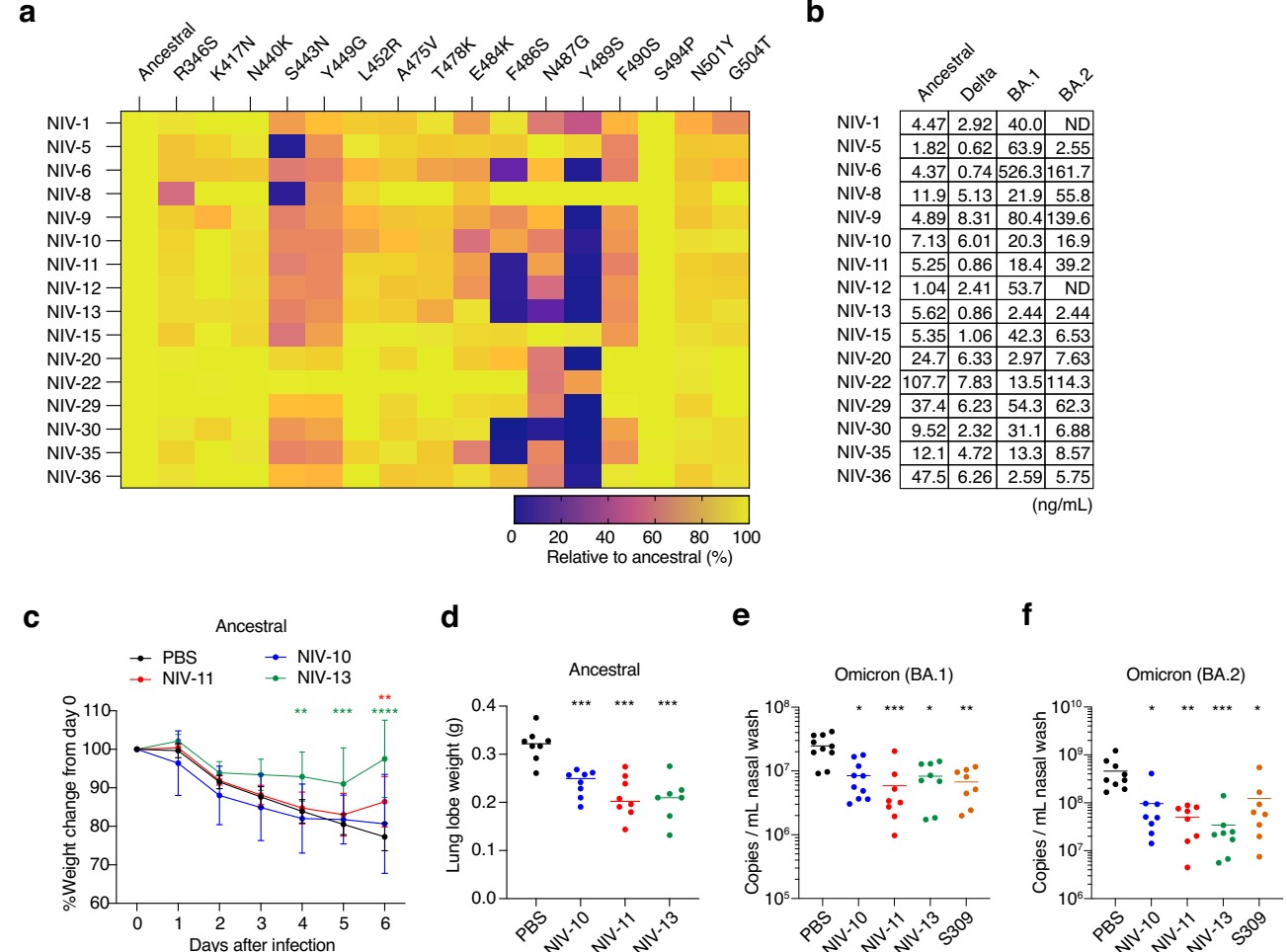

**Fig. 2 | Characterization of broadly neutralizing monoclonal antibodies.**
**a** Binding profiles of each recombinant IgG1 clone to the corresponding RBD mutants were examined by mesoscale, and percentages relative to ancestral RBD are shown in color. **b** Neutralization of each recombinant antibody was examined with an authentic virus in vitro, and $IC_{50}$ values are shown. ND, antibody without a detectable level of neutralization in the experimental condition. **c, d** In vivo treatment effect of indicated antibodies to Syrian hamsters infected with an ancestral virus. Syrian hamsters were intranasally inoculated with $10^4$ $TCID_{50}$ virus, and 5 mg/kg of each monoclonal antibody was administered intraperitoneally on the next day. Body weight change and weight of lung lobe on day 6 are shown. **e, f** In vivo treatment effect of indicated antibodies to Syrian hamsters infected with BA.1

(**e**) and BA.2 (**f**). Syrian hamsters were intranasally inoculated with $10^4$ $TCID_{50}$ virus, and 5 mg/kg of each monoclonal antibody was administered intraperitoneally on the next day. Subgenomic viral RNA copy numbers in the nasal wash on day 3 are shown. Data shown in **a, b** are representative of more than two independent experiments. Data shown in **c–f** are from two independent experiments. Data are presented as mean value ± SD ($n = 8$ biological replicate, **c**). Each symbol represents data from one hamster ($n = 7$–10 biological replicate, **d–f**), and bars represent the mean value (**d–f**). $*p = <0.05$ (**e, f**); $**p = 0.004$ (**c**, NIV-13), $p = 0.0034$ (**c**, NIV-11), $p < 0.01$ (**e, f**); $***p = 0.0006$ (**c**), $p < 0.001$ (**d–f**); $****p < 0.0001$ (two-way ANOVA with Dunnett's test for body weight change, Kruskal–Wallis test for lung weight and qPCR).

antigenic distance at the time of analysis and an authentic RBD probe was used for single-cell sorting. After culturing on feeder cells, single B-cell culture supernatants were screened by enzyme-linked immunosorbent assay (ELISA) using RBD mutant panels with low ACE2 binding (Supplementary Fig. 1c). The F486 mutation was additionally included because it is a representative RBD mutation in the Omicron subvariants BA.4/5, which emerged after BA.2. More than 85% of the clones were bound to both ancestral and Beta RBDs, among which 12.5% lost binding to Y489S (Fig. 1c). These data suggested a substantial contribution of Y489 to the antibody footprint. S443N non-binders (S443N-) were the second-largest population (3.8%), followed by F486S non-binders (F486S−) (3.6%). Neutralization assays using authentic viruses were performed to assess the neutralizing breadth of the individual clones (Fig. 1c and Supplementary Fig. 1d). Of note, the frequency of Y489S non-binders (Y489S−) dramatically increased to 80% among broadly neutralizing antibodies against the ancestral Wuhan strain and variants (Beta, Delta, and BA.1 Omicron). Thus, Y489 is a key component of variant-neutralizing and highly potent

neutralizing antibodies[14,15]. The dominance of Y489S− depended on the breadth of four variants rather than the activity of any particular variant (Supplementary Fig. 1d). These data suggest that Y489 is a critical site for the broad neutralizing activity against emerging variants, including Omicron. The diminished infectivity of the pseudotyped virus with the Y489S mutation also strengthened the idea that this is a site of vulnerability (Supplementary Fig. 1e). F486 is an additional important site; most neutralizing clones recognize Y489 only, and some recognize both sites (Fig. 1d).

For further analysis, sixteen clones were selected as binders to Y489, F486, or S443. The clones were examined using a multiplex panel of single amino acid RBD mutants (Fig. 2a). The antibodies can be divided into three groups; Y489S−F486S− (NIV-6, −11, −12, −13, −30, −35), Y489S−F486S+ (NIV-1, −9, −10, −20, −22, −29, −36), and S443N- (NIV-5, −8, −15). All recombinant antibodies showed high neutralizing ability against the ancestral SARS-CoV-2 and Delta variants (Fig. 2b). Of note, three (NIV-11, −12, −35) out of six Y489S−F486S− clones shared the same clonotype (IGHV1-58/IGHJ3 and IGKV3-20/IGKJ1), which is

frequently utilized in BA.1/BA.2-neutralizing antibodies from diverse individuals (Supplementary Table 2)[8,16]. Furthermore, among the Y489S−F486S+ group, three clones (NIV-20, −29, and −36) utilized the public clonotypes (IGHV3-53/IGKV3-15 and IGHV3-66/IGKV3-15), whose binding modes and neutralizing potency/breadth have been analyzed in previous studies[7,17], owing to the convergence and publicity of these clonotypes in neutralizing antibodies.

Next, we examined the therapeutic activity of NIV-10 (Y489S−F486S+), NIV-11, and NIV-13 (Y489S−F486S−) in vivo using a Syrian hamster model (Fig. 2c–f). These antibodies had a high affinity against ancestral and Delta variant RBDs at the pM and nM orders for BA.1 and BA.2 Omicron RBDs (Supplementary Table 3). Ancestral virus-infected hamsters treated with PBS lost weight from day 0 to 6, but hamsters treated with each neutralizing antibody recovered at later time points. In addition, lung lobe weights were reduced in the antibody-treated group, suggesting attenuated inflammation. For the Omicron challenge experiment, hamsters were infected with the BA.1 or BA.2 variant and treated with antibodies the following day. We measured the viral load in the nasal wash on day 3 due to the low pathogenicity of the variants in the hamster model[18]. Antibody-treated animals had a lower viral load in the nasal wash than in the PBS-treated group, and the copy number was comparable to that in the group treated with S309, a parent antibody of sotrovimab with strong therapeutic activity in the animal model (Fig. 2e, f)[19]. Together, NIV-10, −11, and −13 recognize virus-vulnerable Y489 sites and have cross-neutralizing capacities against Omicron BA.1 and BA.2 variants in vitro and in vivo.

## Antibody of Y489S−F486S+ group is highly resistant to escape mutation

We next analyzed the antibody susceptibility to viral escape using two approaches. In the first approach, deep mutational scanning (DMS) of the RBD was performed to comprehensively evaluate the escapability due to single amino acid substitutions. We employed an inverted infection assay where ACE2-harboring viruses infected spike-expressing cells[20]. This approach can directly analyze the viral escape reflecting both alterations in antibody binding and infectivity. The DMS library based on the ancestral strain spike protein was expressed in human Expi293F cells, followed by pre-treatment with antibodies and incubation with ACE2-harboring GFP reporter viruses (Fig. 3a, b; and Supplementary Fig. 2a). NIV-8 was added as a reference antibody that broadly neutralized the variants but did not recognize Y489. NIV-8 escaped by mutations at sites 346, 443–449, and 499, which is consistent with the binding profile of RBD mutants (Fig. 2a). NIV-11 and NIV-13 had similar escape profiles limited to positions 486 and 487, reproducing the reduced binding to F486S and N487G (Fig. 2a). NIV-10 exhibited overall low escape profiles compared to those of the other three antibodies, with the highest escape fraction observed at site 485 from G to P. A neutralization assay with pseudoviruses harboring the top five NIV-10 escape value substitutions, which do not occur by single nucleotide mutations (Supplementary Table 4), confirmed the DMS data, showing that NIV-10 was highly resistant to escape mutations other than G485P (Fig. 3c, d). Molecular dynamics (MD) simulations of ancestral RBD and the G485P mutant suggested that the position at 485 was one of the most flexible sites in the RBD, and the G485P mutation altered the dynamics of the ACE2-binding site on the RBD (Supplementary Fig. 2b). Upon mutating G485, positively correlated motions between the residues at 485 and around 472–476 disappeared by restraining the backbone angles by the proline ring (Supplementary Fig. 2c), which would hamper the binding and neutralization of the antibody. Notably, the G485P mutation has been found in only three viral genomes out of >15 million sequences deposited in the GISAID database as of May 1, 2023.

Under the second approach, we passaged an authentic virus in vitro multiple times in the presence of serially diluted antibodies to test the antibody resistance to viral escape. Then, we analyzed the RBD

sequence to observe whether any viral escape mutants emerged[10]. As expected from the binding profile toward RBD mutants and DMS analysis, 84.8% of NIV-8-escape mutants had K444M, and 14.5% had G447D (Supplementary Fig. 2d). Likewise, the NIV-13-escape mutants had F486S in all sequences. In contrast, no single mutations were found in the RBD of the viruses under the same passage conditions in NIV-10. Mutants were not found in NIV-11 for unknown reasons. Thus, among Y489S−F486S+ group, we identified the RBS antibody (NIV-10) exhibiting extreme resistance to the emergence of escape mutations by two independent approaches.

## Y489S−F486S+ antibody group retained neutralizing activity to BA.4/5

Omicron subvariants BA.4 and BA.5 carry point mutations that change F486 to V. Because most broadly neutralizing antibodies against variants recognize F486 and are encoded by public clonotypes (e.g., IGHV1-58), the F486 mutation in BA.4/5 variants could emerge as a result of selective pressure by variant-neutralizing antibodies as part of the herd immunity. Consistent with this scenario, NIV-11 (IGHV1-58) and NIV-13 of the Y489S−F486S− group reduced the neutralizing activity of pseudotyped viruses carrying BA.4/5 mutations (Fig. 4a). The evasion was reproduced by the authentic BA.5 virus strain; the Y489S−F486S− group failed to neutralize the BA.5 virus, but the Y489S−F486S+ group, including NIV-10, retained neutralizing activity (Fig. 4b, c). We also examined neutralization to BA.2.75, which emerged after BA.4/5, and confirmed that all Y489S− antibodies, except NIV-22, could efficiently neutralize BA.2.75 (Fig. 4d, e).

The RBS harbors F486, and mutations in this residue interfere with ACE2 binding[9]. Indeed, the F486S− mutant strain selected by NIV-13 showed a reduced ability to replicate in the VeroE6-TMPRSS2 cell line compared to the parental ancestral strain after the same passage without antibody selection (Fig. 4f). In contrast, the F486 mutant strain selected from the BA.1 virus exhibited replication comparable to that of the parent BA.1 strain (Fig. 4g), supporting the circulation of the F486-mutated Omicron subvariant in the real world. This result is consistent with the epistatic effects of other Omicron mutations[21]. These results suggest that the F486S mutation negatively affects viral replication owing to reduced ACE2 binding, but the Omicron BA.1 mutation could circumvent this effect. Similarly, the amino acid at position 486 was changed from serine to proline in the XBB.1.5 sublineage, resulting in a marked increase in affinity for ACE2[22].

We also examined the effects of F486 mutations on plasma IgG levels. IgG titers against ancestral, F486S, and Y489S RBDs in vaccine plasma were measured by ELISA 1 month after the 2nd vaccination (T5), 5–6 months after the 2nd vaccination (T6), and 1 month after the 3rd vaccination (T7) (Fig. 4h). The F486S mutation significantly reduced anti-RBD IgG at all time points examined. The difference was more profound in T5 than in T6 and T7 (0.28-times vs. 0.49-times and 0.51-times). We subsequently examined the neutralizing activity of T7 plasma against BA.1, BA.1 + F486S, BA.2, and BA.5 (Fig. 4i, j). T5 and T6 plasma were not applicable to this experiment, as their BA.1 authentic virus-neutralizing activity was undetectable. Consistent with the ELISA data, the neutralizing activity of T7 plasma against both, BA.1 + F486S and BA.5 were 0.46-times lower compared to BA.1 and BA.2, respectively. Thus, F486 is one of the key residues in the Omicron variant recognized by plasma-neutralizing antibodies in vaccines, and potentially provides selective pressure for escape mutations.

## NIV-10 recognizes a broad epitope on the RBD

The structures of the SARS-CoV-2 spike protein complexes with NIV-8, NIV-10, NIV-11, and NIV-13 Fab were determined using cryo-electron microscopy (cryo-EM) (Fig. 5a, Supplementary Figs. 3–6, Supplementary Tables 5 and 6). As the density of the RBD-NIV-10 interface was unclear from the cryo-EM analysis because of the flexibility of the spike, the epitope of NIV-10 was identified using the structure of the

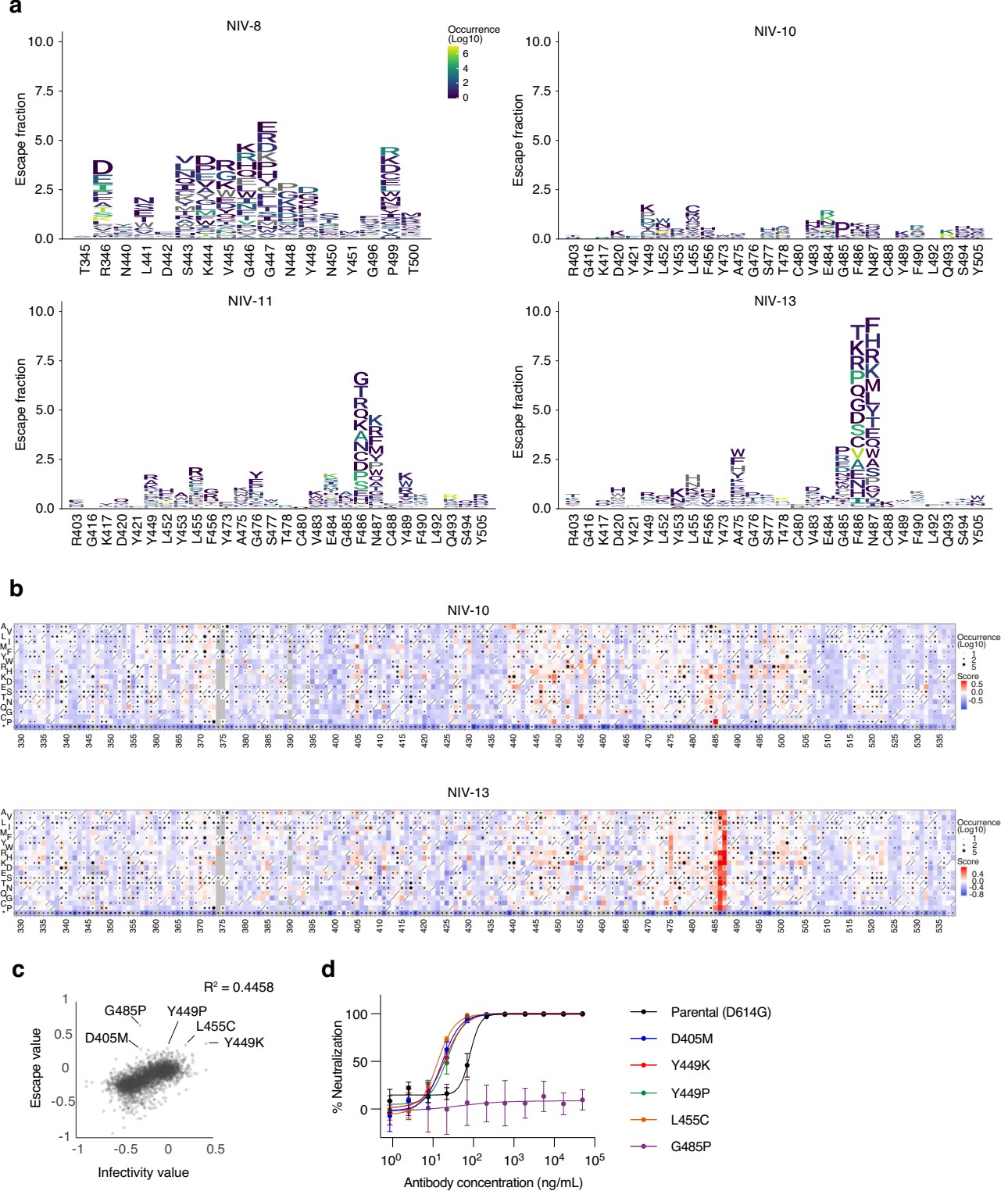

**Fig. 3 | Escape mutation analysis of representative antibodies. a** The effect of a single amino acid mutation in RBD on antibody binding was examined by DMS in the context of ACE2-harboring virus infection to human Expi293F cells expressing the full-length spike protein. The region of epitopes for NIV-8 or NIV-10, −11, and −13 is shown. Amino acid substitutions with higher natural occurrence numbers are shown in yellow, and lower occurrence numbers are shown in navy. **b** Full heat maps of escape fractions for NIV-10 and NIV-13 are shown. See the supplemental figure for other heat maps. Squares are colored by substitution effect according to the scale bar on the right. Squares with a diagonal line through them indicate the ancestral strain amino acid. Black dot size reflects the frequency in the virus genome sequence according to the GISAID database as of 7 July 2022. **c** The correlation in effects of single amino acid substitutions on the alteration of infectivity and escapability from NIV-10 is shown. DMS data for infectivity is retrieved from Ikemura et al.[35] **d** Neutralization of NIV-10 against pseudoviruses harboring the top 5 escape value substitutions and parental D614G mutation are shown. Data are representative of two independent experiments and are presented as mean value ± SD (n = 4, technical replicate).

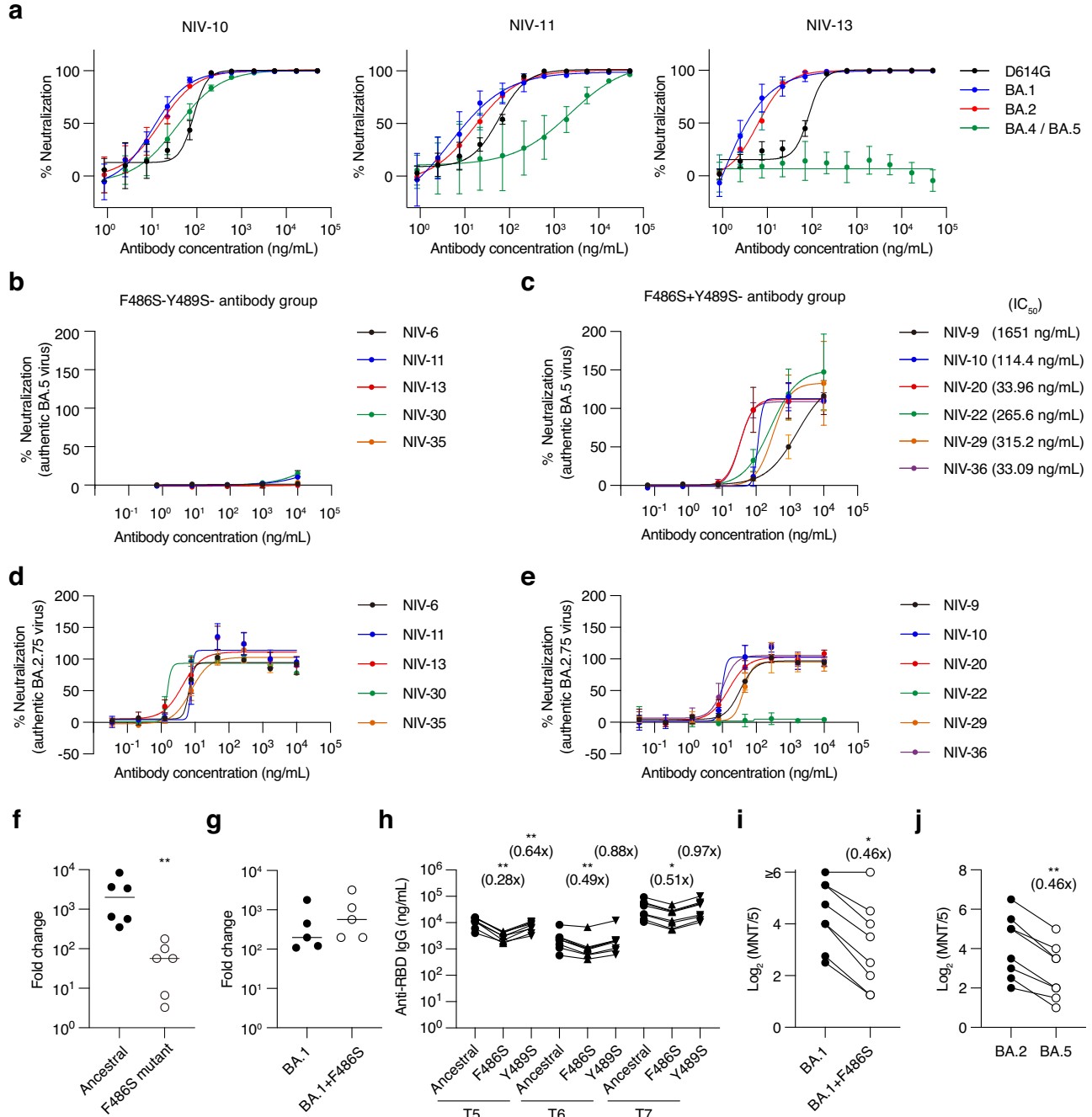

**Fig. 4 | Effect of F486S mutation on antibody binding and neutralization.**
**a** Neutralization of NIV-10, NIV-11, and NIV-13 was examined with pseudovirus containing indicated spike protein. Data are presented as mean value ± SD ($n = 4$, technical replicate). **b, c** Neutralization of indicated antibodies and IC50 with authentic BA.5 virus are shown. Data are presented as mean value ± SD ($n = 4$, technical replicate). **d, e** Neutralization of indicated antibodies and IC50 with authentic BA.2.75 virus are shown. Data are presented as mean value ± SD ($n = 3$, technical replicate). **f** NIV-13 escape mutant clone and ancestral virus with passage without antibody were infected on VeroE6-TMPRSS2 cells at a multiplicity of infection of 0.01, and TCID50 was examined on days 0 and 3. Fold-change values are shown. Each symbol represents data from each virus clone ($n = 6$, biological replicate). Bars represent the mean value. **g** NIV-13 escape mutant was cloned from BA.1 passaged with NIV-13, and proliferation was examined as **f**. Each symbol represents

data from each virus clone ($n = 5$, biological replicate). Bars represent the mean value. **h** Plasma IgG titer against ancestral and mutant RBDs is shown. Plasma samples were collected from the vaccine 1 month after the 2nd vaccination (T5), 5–6 months after the 2nd vaccination (T6), and 1 month after the 3rd vaccination (T7). Connected symbols represent data from each individual ($n = 8$, biological replicate). **i** Neutralization antibody titer of T7 plasma against BA.1 and BA.1 + F486S viruses are shown. Connected symbols represent data from each individual ($n = 8$, biological replicate). **j** Neutralization antibody titer of T7 plasma against BA.2 and BA.5 viruses are shown. Connected symbols represent data from each individual ($n = 8$, biological replicate). Data shown in (**a–g**) are representative of two independent experiments. *$p < 0.05$ (one-way ANOVA for **h**), $p = 0.0156$ (two-tailed Wilcoxon test for **i**); **$p = 0.0022$ (two-tailed Mann–Whitney test for **f**), $p = 0.0078$ (two-tailed Wilcoxon test for **j**), $p < 0.01$ (one-way ANOVA for **h**).

complex with RBD via X-ray crystallography at 2.2 Å resolution (Supplementary Table 7). The amino acids interacting with Fab (distances within 4.5 Å) were labeled; these are shown as epitope regions of the antibodies (Fig. 5b, c).

The cryo-EM structures show that NIV-8 is a class 3 antibody[23] that binds to RBD in both up and down conformations via the critical S443 footprint. In contrast, NIV-10, NIV-11, and NIV-13 are class 1 antibodies[23] that recognize RBD with epitopes, including F486 and Y489. These epitopes corresponded with the binding profile of each antibody (Fig. 2a). NIV-8 interacted with the RBD (R346, Y449, Y451, and Q498) via CDR-H2 (Y106 and Y107) and CDR-H3 (S52 and Q57) (Supplementary Fig. 7a, left and middle); however, there was no direct interaction with S443, whose substitution with asparagine substantially reduced RBD binding (Supplementary Fig. 7a, right; and Fig. 2a). As the side chain of S443 was inward-facing, the loss of neutralizing activity by the S443N mutation was likely because of the conformational change in the loop containing S443.

On the contrary, the class 1 antibodies NIV-10, NIV-11, and NIV-13 recognized the epitope region, including F486 and Y489; however, the conformations of spike in antibody-bound states were different. NIV-11 recognized only the 3-up conformation of RBDs with C3 symmetry, whereas NIV-10 and NIV-13 bound to both up and down RBDs and showed several conformations (Fig. 5a and Supplementary Fig. 7b). NIV-10 pushed the neighboring RBD outward and then bound to the down RBD and RBD in intermediate states to avoid a conflict with the adjacent RBD. NIV-13 bound to the down state without interfering with the neighboring down RBD, possibly because of the narrow recognition region that allowed access to the down RBD state.

NIV-11 belongs to public clonotypes (IGHV1-58 and IGKV3-20) utilized by B1-182.1, S2E12, and UT28K[24–26] which recognize class 1 super sites (Supplementary Fig. 7c, and Supplementary Table 2). The binding mode of NIV-11 was essentially the same as that of IGHV1-58/IGKV3-20 paired antibodies. The side chain of the F486 residue of the RBD interacted with a hydrophobic pocket at the boundary between the CDR-H and CDR-L loops of NIV-11. Furthermore, the side chain of Y29 in the CDR-H1 of NIV-11 formed a hydrogen bond with the side chain of Y421 of the RBD in a unique interaction that depended on the CDR-H1 sequence of NIV-11 (Supplementary Fig. 7c).

Finally, the heavy chain of NIV-10 bound to the neck part of RBD, with the CDR-H3 targeting the left shoulder from the back side (Fig. 5d). L105, located at the tip of CDR-H3, was bound to the hydrophobic pocket formed by L452, F490, and L492. The CDR-H3 of NIV-10 also recognized the area surrounding the F486 of the RBD together with the light chain, which overlapped with the NIV-11 and NIV-13 epitopes (Fig. 5c, e, f). F486 was located close to the hydrophobic environment of the light chain (L46, Y49, and P55). N487 and Y489 formed hydrogen bonds with the heavy chain. Y489 also had π-π stacking interactions with the Y32 and H110 of the heavy chain (Fig. 5e, g). The F486 recognition modes of NIV-10 and NIV-11 were structurally different, and NIV-10 had more space around F486 than NIV-11, possibly accounting for the NIV-10 tolerability to F486 mutations (Fig. 5f). The solvent-accessible surface area (SASA) of F486 was 3.6-times larger in NIV-10 than that in NIV-11 (28.4 Å$^2$ versus 8.0 Å$^2$), whereas F486 in NIV-13 was completely buried (0.0 Å$^2$). To examine the dynamics between F486 and NIV-10 and NIV-13 under aqueous conditions, MD simulations of the RBD in complex with each antibody were performed. The MD simulations revealed the duration of interaction with individual amino acids, highlighting the smaller contributions of F486 to NIV-10 compared to its contributions to NIV-13 (Supplementary Fig. 8a). Changes in the SASA of F486 during the simulations also suggested that F486 had increased exposure to the solvent during its interaction with NIV-10 (Supplementary Fig. 8b). In contrast, F486 was well buried at the V$_L$/V$_H$ interface of NIV-13. These MD simulation data support that F486 is not an essential footprint of NIV-10, thereby permitting neutralization of the BA.4/5 variants.

In contrast, the G485 of the RBD formed hydrogen bonds with the NIV-10 light chain Y49 and heavy chain H110. In addition to eliminating the interaction with these hydrogen bonds, the G485P mutation may cause a conformational change that affects the positions of F486, N487, and Y489, leading to reduced binding with NIV-10 (Fig. 5g). Taken together, NIV-10 had a broader epitope area than NIV-11 and NIV-13, possibly accounting for its resilience to viral escape.

## Y489S−F486S+ antibody group possesses hydrophobic heavy chain CDR3

The NIV-10 footprint includes a hydrophobic region within the RBS (Fig. 6a). Therefore, we compared the hydrophobicity of heavy chain CDR3 between the Y489S−F486S+ antibody group (BA.4/5 neutralizing) and the Y489S−F486S− antibody group (non-neutralizing). Recently reported BA.4/5-neutralizing (Omi-3, −18, −42; BD-515) and non-neutralizing (Omi-2, −25) antibodies were encoded by public clonotypes IGHV3-53 (Omi-3, −18), IGHV1-69 (Omi-2), IGHV3-9 (Omi-25, −42), and IGHV3-66 (BD-515)[27,28], and their hydrophobicity was scored as well (Fig. 6b, c). Remarkably, the hydrophobic scores of BA.4/5-neutralizing antibodies were significantly higher than those of non-neutralizing antibodies (Fig. 6d). Thus, in addition to the broader epitope area, the increased hydrophobicity of this antibody group, especially of NIV-10, which has the highest score, is advantageous for recognizing the hydrophobic region of the RBS. This phenomenon potentially contributes to the BA.4/5-neutralizing activity and high resistance to escape mutations.

## Computationally designed antibody based on NIV-10 can neutralize XBB.1.5

Following the appearance of BA.4/5 and BA.2.75, new Omicron sublineages such as BQ.1.1, XBB, and its sublineage XBB.1.5, have emerged[22,29]. Hydrophobicity calculation of the RBD variants suggested that XBB RBS is less hydrophobic than to other Omicron sublineages, such as BA.5 and BQ.1.1. Moreover, XBB.1.5 RBS is slightly more hydrophobic than XBB, due to the proline mutation at position 486 (Supplementary Fig. 9). We further examined whether NIV-10 was resistant to these new variants and found that NIV-10 potently neutralized BQ.1.1, but reduced the activity to XBB. Furthermore, NIV-10 did not neutralize XBB.1.5 even at 100 µg/mL (Fig. 7a, b).

Because XBB.1.5 has a new substitution at position 486 (F486P), we investigated the effect of the amino-acid substitution at this position on antibody binding. We expressed recombinant XBB RBD with a single amino acid substitution at position 486, which can occur by a single nucleotide mutation in F486 and S486. The C486 mutant was not properly expressed and was not included in this analysis. The binding of NIV-10 to the original XBB RBD (S486) and XBB RBD with A486, Y486, T486, L486, I486, V486, R486, and P486 mutations, was examined by ECLIA (Fig. 7c). We found that the SAP value and binding signal of each XBB RBD mutant correlated well with the antibodies, except for the P486 mutant (namely XBB.1.5). We also analyzed the Omi-3 and XBB RBD mutants and found that the SAP value and binding signal were strongly correlated, even for the P486 mutant. These results demonstrated that proline substitution at position 486 specifically inhibits NIV-10 binding and abolishes the XBB.1.5-neutralizating activity. Consistent with these findings, structural modeling suggested that the compatibility of NIV-10 with XBB.1.5 was impeded by a proline mutation at position 486 (Supplementary Fig. 10a). When the amino acid sequence of XBB.1.5 RBD was computationally mapped onto the crystal structure of the NIV-10/Ancestral RBD complex, we observed a steric clash, specifically at position 486 of the interface. Docking simulations also indicated that even after refinement calculations to alleviate the steric clash at the interface, greater structural variation was observed in the NIV-10/XBB.1.5 RBD complex (Supplementary Fig. 10b). The interface energy for this complex was the poorest, followed by that of the NIV-10/XBB RBD and NIV-10/ancestral RBD

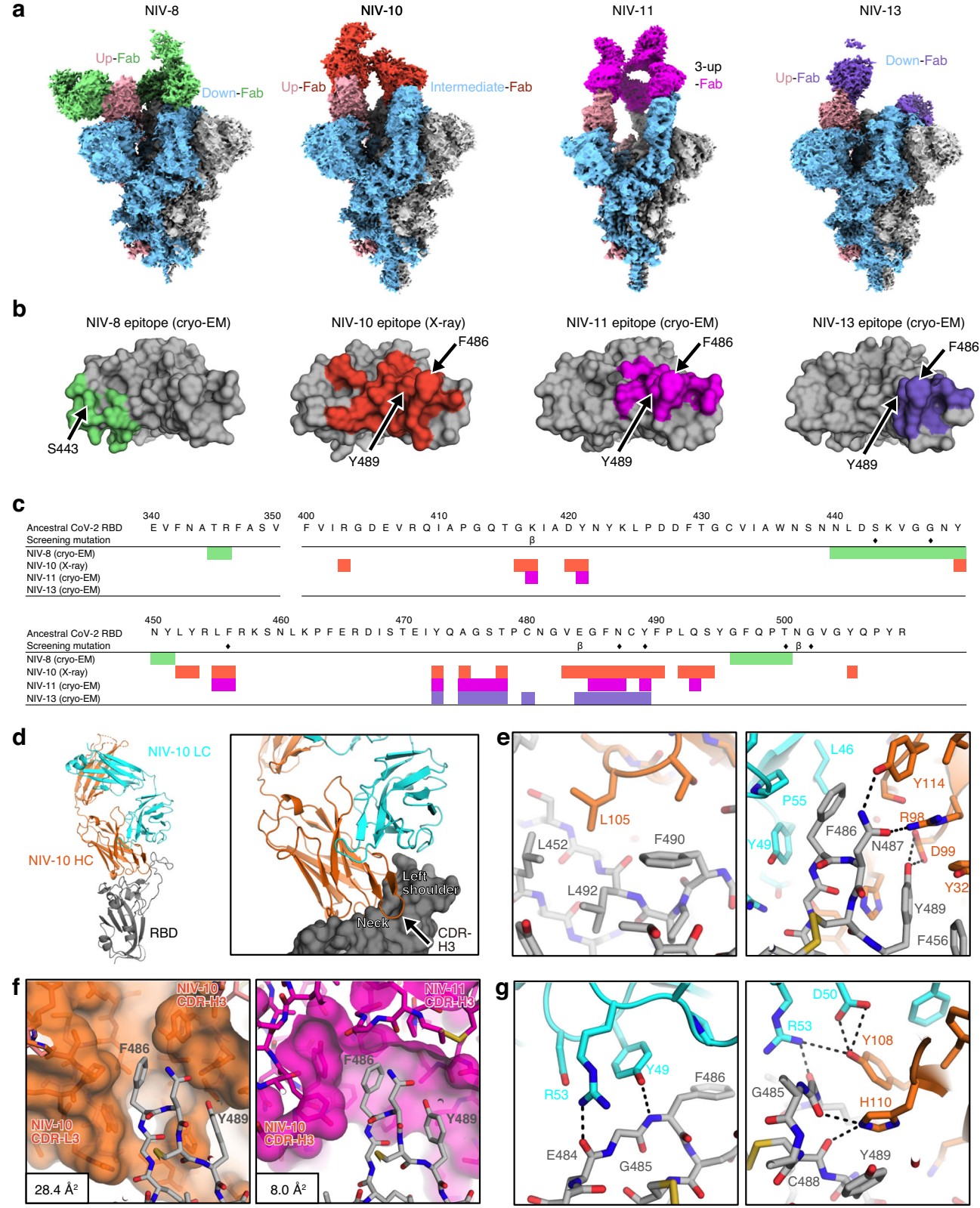

complexes (Supplementary Fig. 10c). Collectively, these results suggest that the interface of NIV-10/XBB.1.5 RBD was the most physiochemically and sterically incompatible.

To overcome the limitation of NIV-10, we utilized a computational design to create a new antibody sequence based on NIV-10, focusing on the interaction energy, shape complementarity, and the number of hydrogen bonds at the antibody-antigen interface. This approach led

to the rational design of an NIV-10-derivative antibody, NIV-10/FD03, which exhibited a compatible interface with XBB.1.5 (Supplementary Fig. 10a) and a more hydrophobic paratope compared to the parental NIV-10 (Supplementary Fig. 10d). Recombinant NIV-10/FD03 exhibited strong binding to XBB RBDs, irrespective of mutations at position 486 (Fig. 7d). Furthermore, NIV-10/FD03 antibody potently bound to various variant RBDs (Fig. 7e). Finally, we examined the neutralizing

**Fig. 5 | Structural analysis of ancestral spike trimer protein and NIV−8, −10, −11, and −13 Fabs. a** Representative cryo-EM global maps of each Fab in complex with the ancestral spike protein. NIV-8, −10, −11, and −13 are shown in green, orange, magenta, and purple, respectively. The colors of the map of the spike protein were assigned to gray, sky blue, and pink for each protomer. **b** RBD is shown from the top view. The epitope of NIV-8, NIV-10, and NIV-11 Fabs are colored for residues within 4.5 Å, while that of NIV-13 Fab is colored for residues within 6 Å of main-chain atoms of Fab because of limited resolution structures (~4 Å). Arrows indicate S443, F486, and Y489. **c** Residues highlighted in **b** are mapped for NIV-8, −10, −11, and NIV-13. Epitope residues for each antibody are represented in the same color as **b**; seven

amino-acid mutations selected for screening are marked with a diamond. The mutations in the Beta-variant are shown as β under the RBD sequence. **d** Overview of Spike-NIV-10 complex (Spike; gray, NIV-10; orange (heavy chain), cyan (light chain)) (left) and the close-up view of its interface (right). **e, g** Detailed recognition mode of NIV-10 toward the region surrounding F486 and Y489 (**e**) and G485 (**g**) of RBD. The dotted lines are indicated as polar interactions. **f** Structural comparison of recognition of NIV-10 (left) and NIV-11 (right) with F486 and Y489 of RBD. The surfaces of NIV-10 and NIV-11 are also shown. Solvent-accessible surface area (SASA) of RBD F486 is calculated and shown in the left bottom box.

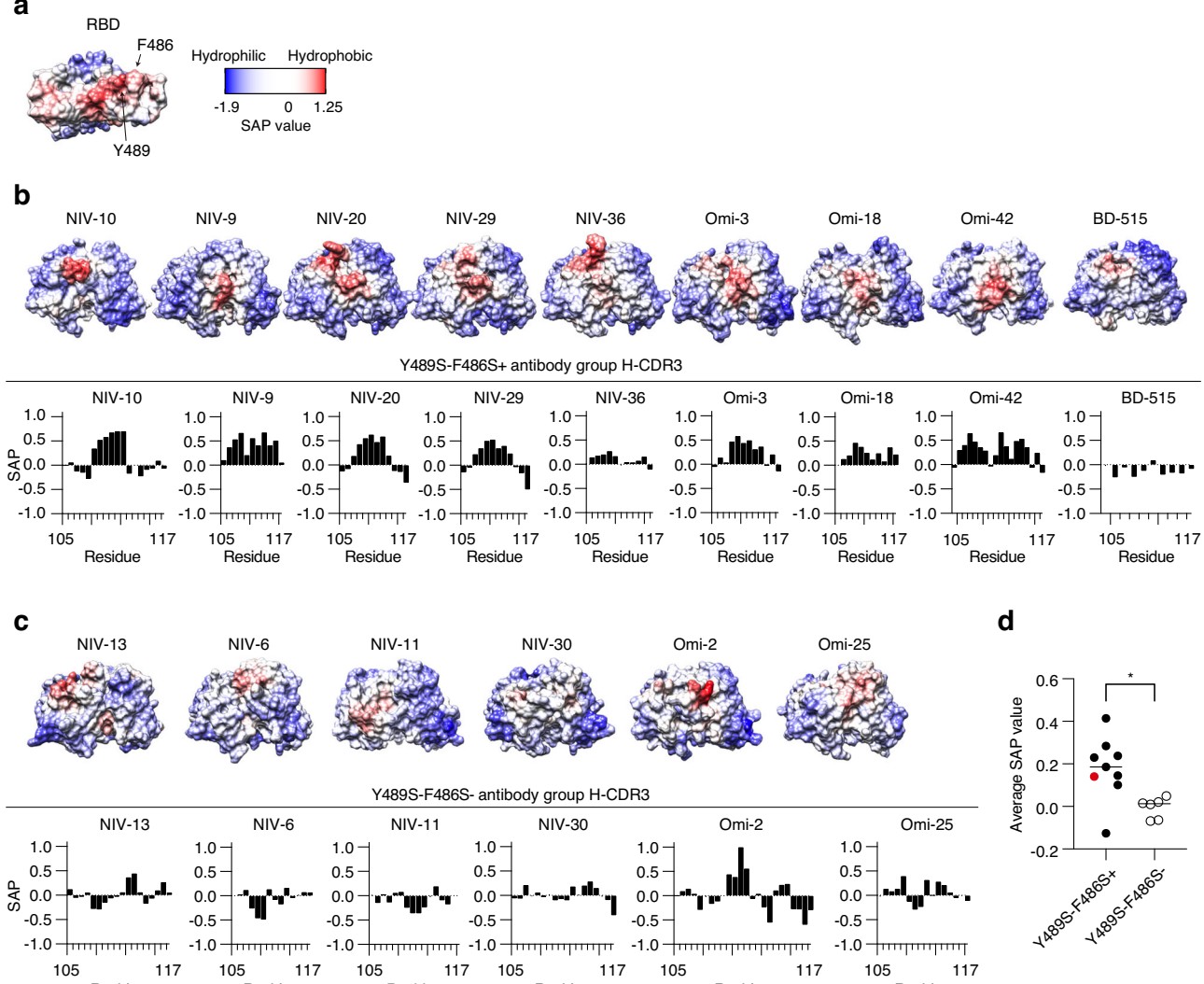

**Fig. 6 | Hydrophobicity of RBD and H-CDR3 of monoclonal antibodies. a** Spatial aggregation propensity (SAP) values for ancestral RBD are presented as graphics. **b** SAP value for each variable fragment and heavy chain CDR3 (residue 105 to 117 in the IMGT numbering) of the Y489S−F486S+ antibody group are presented as graphics and graphs. **c** SAP value for each variable fragment and heavy chain CDR3 of the Y489S−F486S− antibody group, are presented as graphics and graphs.

**d** Averaged SAP values for heavy chain CDR3 of Y489S−F486S+ antibody group (n = 9, biological replicate) and Y489S−F486S− antibody group (n = 6, biological replicate) are shown. Each symbol represents data from antibody. The SAP value of NIV-10 is highlighted in red. Bars represent the mean value. *p = 0.012 (two-tailed Mann–Whitney test).

abilities of NIV-10/FD03 and found that the designed antibody effectively neutralized XBB and XBB.1.5 (IC₅₀ = 56.78 ng/mL) (Fig. 7f).

## Discussion

Since the emergence of SARS-CoV-2, many neutralizing antibodies have been isolated for therapeutic application. However, a limited number of antibodies remained resistant to emerging Omicron

subvariants due to the emergence of SARS-CoV-2 variants with amino acid substitutions in the key footprints. Here, we applied a rational screening approach to screen neutralizing antibodies with extreme mutation-resilience, and identified three groups of antibodies from COVID-19 convalescent PBMCs; Y489S−F486S− (NIV-6, −11, −12, −13, −30, −35), Y489S−F486S+ (NIV-1, −9, −10, −20, −22, −29, −36), and S443N- (NIV-5, −8, −15). Most antibodies showed potent neutralization

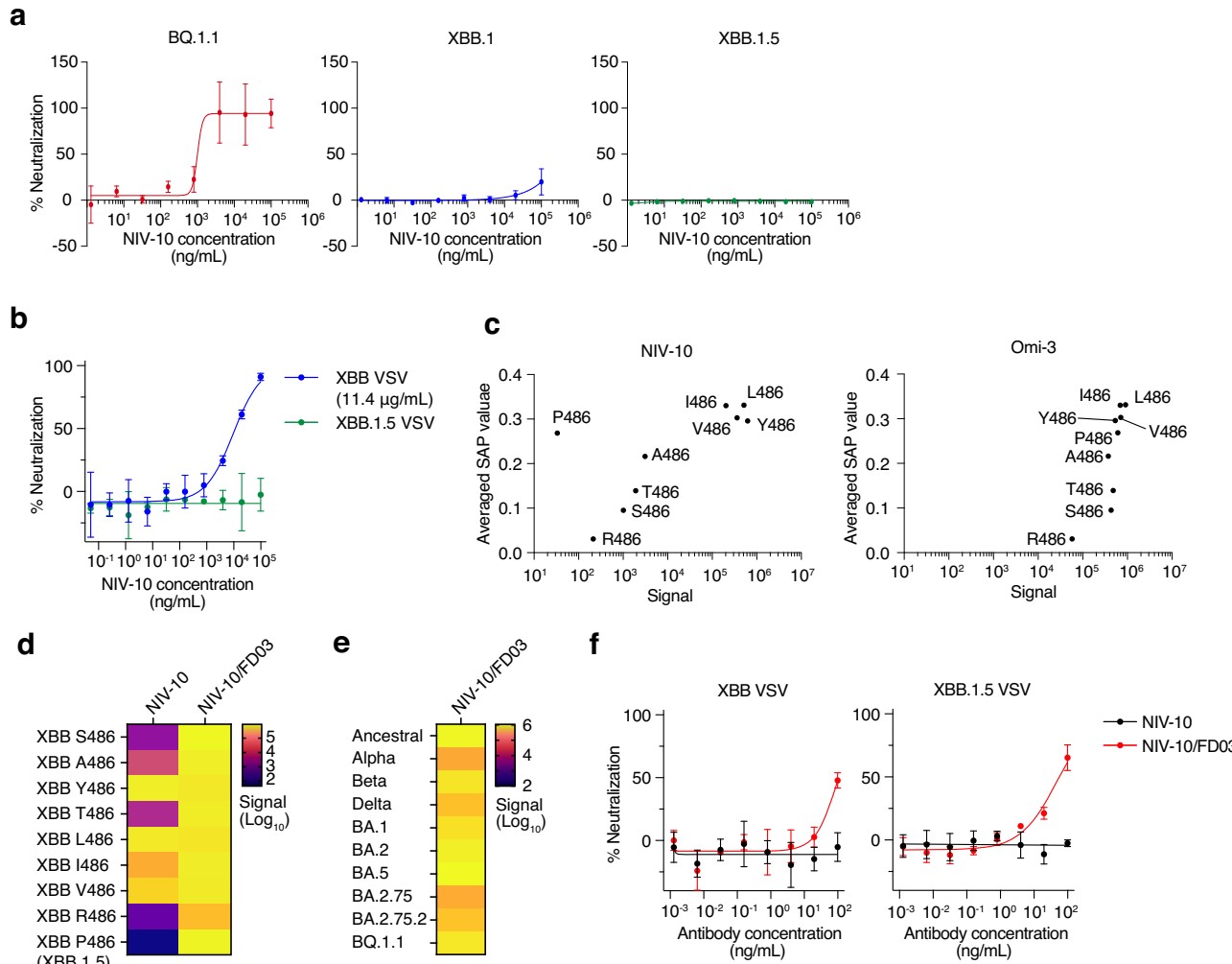

**Fig. 7 | Binding and neutralization of XBB.1 and XBB.1.5 by NIV-10 and computationally designed antibody NIV-10/FD03. a, b** Neutralization of NIV-10 was examined with authentic viruses (**a**) and a pseudovirus containing XBB.1.5 spike protein (**b**). IC$_{50}$ with XBB VSV was shown in brackets. Data are presented as the mean value ± SD (*n* = 4 for **a** and 3 for **b**, technical replicate). **c** Averaged SAP values of XBB RBDs with the indicated mutations at 486 and their binding signal to NIV-10 and Omi-3 are plotted. **d** Binding signals of XBB RBDs with indicated mutations at 486, NIV-10, and NIV-10/FD03 are shown as a heatmap. **e** Binding signals of the indicated RBDs and NIV-10/FD03 are shown as a heatmap. **f** Neutralization of NIV-10 and NIV-10/FD03 with pseudovirus containing XBB and XBB.1.5 spike protein. Data are presented as the mean value ± SD (*n* = 3, technical replicate). The data shown in (**a, b, d, f**) are representative of two independent experiments. The data shown in (**c, e**) are the averages of two independent experiments.

to the ancestral strain, Delta variant, and Omicron subvariants, such as BA.1, BA.2, and BA.2.75. Y489S−F486S− antibodies were susceptible to BA.4/5, which possesses an F486V substitution. One caveat in our escape estimation is that we potentially overlooked the influence of epistatic effects by variant lineage-related mutations, which was shown in a previous study[21].

Based on the structural data, the recognition pattern of SARS-CoV-2 RBD antibodies has been subgrouped in great detail. Highly potent ACE2-blocking antibodies are mainly mapped to the neck and left shoulder of the RBD[30]. Cryo-EM data mapped the epitopes of NIV-11 and NIV-13 to the left shoulder of the RBD, similar to the potent broadly neutralizing antibodies B1-182.1, S2E12, and UT28K[24−26]. F486 is positioned at the top of the left shoulder and is critical for binding these antibodies. Therefore, an F486V mutation in BA.4/5 subvariants permits the escape from this group of potent neutralizing antibodies.

In contrast, NIV-10 was mapped to the neck to the left shoulder RBS region of the RBD, partially overlapped with NIV-13, but recognized a relatively broad epitope including F486. Of note, NIV-10 was resistant to F486 substitution and was able to neutralize BA.4/5. DMS data demonstrated overall high resistance of NIV-10 to an amino acid

substitution in the RBD, and G485P was the highest escape substitution for NIV-10. These data suggest that NIV-10-spike binding was maintained by multiple complementary interactions instead of a few critical interactions. Concordantly, MD simulation data suggest the presence of highly interacting amino acids in the epitope of NIV-13 but fewer highly interacting amino acids in the epitope of NIV-10. We found that NIV-10 and other Y489S−F486S+ antibodies had a hydrophobic H-CDR3, whereas H-CDR3 of Y489S−F486S− antibodies, including NIV-13, were less hydrophobic. Hydrophobic H-CDR3 of NIV-10 and other Y489S−F486S+ antibodies may be beneficial for the interaction with the neck to the left shoulder RBS region of the RBD, which is hydrophobic.

To our surprise, 80% of COVID-19 convalescent-derived monoclonal antibodies with broad neutralizing activity against Beta, Delta, and BA.1 variant recognized Y489 in the RBS. Our serological data showed the presence of a significant amount of Y489-recognizing antibodies in the plasma of individuals one month after the 2nd vaccination. However, we did not observe a significant inclusion of Y489-recognizing antibodies at 5−6 months after the 2nd vaccination and 1 month after the 3rd vaccination. Y489S− memory B cells should be

analyzed longitudinally in a future study. Among the broadly neutralizing Y489S−F486S+ group, NIV-20, −29, and −36 utilized the public clonotypes (IGHV3-53/IGKV3-15 and IGHV3-66/IGKV3-15). This antibody group with public clonotypes could be a new target for broadly protective vaccine design for further boosting. In contrast, F486-recognizing antibodies were present in the plasma of individuals 5−6 months after the 2nd vaccination and 1 month after the 3rd vaccination, as well as one month after the 2nd vaccination. These data suggest the elicitation and recall of F486-recognizing antibodies/B cells by antigen exposure. Therefore, these antibodies may have worked as a selective pressure for viral escapes with F486 substitution.

NIV-10 potently neutralized one of the recent Omicron subvariants, BQ.1.1; however, it retained only mild neutralizing activity against XBB.1, which might be attributed to the reduced hydrophobicity of the NIV-10 epitope on XBB.1 RBD compared to that of BQ.1.1 and other variant RBDs. The XBB.1.5 subvariant partially restored epitope hydrophobicity by replacing the P486 mutation; nonetheless, it escaped from NIV-10 neutralization to a greater extent than XBB.1. The binding of NIV-10 and XBB RBDs with an amino acid substitution at position 486 correlated well with the hydrophobicity of RBD, except for proline substitution, which specifically impeded NIV-10 binding to XBB.1.5. Based on computational modeling, we speculated that the F486P mutation creates a steric clash. The intrinsic structural dynamics of NIV-10, which may remove such a clash, could result in further incompatibility at the antibody-antigen interface.

To improve the attenuated neutralizing activity of NIV-10 against XBB sublineages, we utilized a computational design and successfully generated an engineered NIV-10-derivative antibody, NIV-10/FD03. We verified that NIV-10/FD03 could potently bind to RBDs with various mutations, including XBB.1.5 with the neutralizing activity. Currently, only a few antibody therapeutic candidates are effective against XBB.1.5[22]. As demonstrated in this study, Y489S−F486S+ antibodies may serve as reasonable templates for designing new and effective antibody therapeutics.

Collectively, we described a group of RBS-binding antibodies that broadly neutralize the Omicron variant, including the BA.4/5 subvariant with the F486V mutation. Among the Y489S−F486S+ antibody group, NIV-10 recognizes a broader epitope, likely due to its hydrophobicity in the antigen-binding domain, and possesses high resistance to escape mutations. Considering the rarity of the NIV-10 clonotype in herd immunity, it is reasonable to speculate that the emergence of the NIV-10-escape mutation is rare. Y489S−F486S+ antibodies may be a promising antibody target that should be boosted by broadly protective vaccines because some of the Y489S−F486S+ antibodies are encoded by public clonotypes and may be widely preserved in those who acquired immunity via previous infection or vaccination.

## Methods

### Protein production
Recombinant proteins were produced as described previously[31]. Briefly, human codon-optimized SARS-CoV-2 spike protein amino acid 331-529, N-terminal signal peptide sequence (MIHSVFLLMFLLTPTESYVD), and C-terminal His-tag and Avi-tag were cloned into the pCAGGS vector. Following mutations were introduced for producing variant RBDs; N501Y for Alpha, K417N / E484K / N501Y for Beta, L452R / T478K for Delta, G339D / S371L / S373P / S375F / K417N / N440K / G446S / S477N / T478K / E484A / Q493R / G496S / Q498R / N501Y / Y505H for BA.1, G504D / S371F / S375F / T376A / D405N / R408S / K417N / N440K / S477N / T478K / E484A / Q493R / Q498R / N501Y / Y505H for BA.2, G339D / S371F / S373P / S375F / T376A / D405N / R408S / K417N / N440K / G446S / N460K / S477N / T478K / E484A / Q498R / N501Y / Y505H for BA.2.75, G339D / R346T / S371F / S373P / S375F / T376A / D405N / R408S / K417N / N440K / G446S / N460K / S477N / T478K / E484A / F486S / Q498R /

N501Y / Y505H for BA.2.75.2, G339D / S371F / S373P / S375F / T376A / D405N / R408S / K417N / N440K / L452R / S477N / T478K / E484A / F486V / Q498R / N501Y / Y505H for BA.5, G339D / R346T / S371F / S373P / S375F / T376A / D405N / R408S / K417N / N440K / K444T / L452R / N460K / S477N / T478K / E484A / F486V / Q498R / N501Y / Y505H for BQ.1.1, G339D / R346T / L368I / S371F / S373P / S375F / T376A / D405N / R408S / K417N / N440K / V445P / G446S / N460K / S477N / T478K / E484A / F486S / F490S / Q498R / N501Y / Y505H for XBB, G339H / R346T / L368I / S371F / S373P / S375F / T376A / D405N / R408S / K417N / N440K / V445P / G446S / N460K / S477N / T478K / E484A / F486P / F490S / Q498R / N501Y / Y505H for XBB.1.5. The RBD expression vector was expressed using Expi293 (Thermo Fisher Scientific) and purified using a TALON column (Clontech). For some experiments, the RBD expression vector and BirA-Flag plasmid (Addgene) were co-expressed, and 100 μM biotin was added to biotinylate the RBDs. For crystallization, RBD protein was prepared and deglycosylated as follows; the RBD expression vector was expressed in HEK293T cells (a human embryonic kidney cell line; ATCC, CRL-3216) in the presence of Kifunensine (funakoshi). The supernatant from cultured cells was collected and RBD protein was purified by Ni-NTA followed by size exclusion chromatography equilibrated with 20 mM Tris-HCl pH8.0 and 100 mM NaCl. After purification, RBD was treated with endoglycosidase H (EndoHf) (NEB) for deglycosylation. EndoHf was removed by size exclusion chromatography in the same condition above.

Recombinant monoclonal antibodies and Fabs were produced as described previously[10,32,33]. Briefly, variable regions of the immunoglobulin heavy chain and light chain from single-cell cultures or published monoclonal antibodies were cloned into human IgG1 heavy chain, heavy chain CH1, kappa, or lambda light chain expression vectors. Pairs of heavy chain and light chain vectors were co-expressed on the Expi293 system (Thermo Fisher Scientific), IgG1 was purified with a protein G column (Thermo Fisher Scientific), and Fab was purified using a Talon column (Clontech).

### PBMC preparation
SARS-CoV-2-infected individuals were enrolled at the Tokyo Center Clinic. Blood samples were collected in Vacutainer CPT tubes (BD Biosciences), and PBMC and plasma samples were isolated via centrifugation according to the manufacturer's instructions. All convalescent samples were seropositive for nucleocapsid antibodies using Elecsys Anti-SARS-CoV-2 (Roche). Vaccinated individuals were enrolled at the Tokyo Metropolitan Bokutoh Hospital, and blood samples were collected longitudinally at 27−34 days (T5) and 146−154 days (T6) after 2nd vaccination and 1 month after 3rd vaccination (T7). All studies were approved by the institutional review board of the National Institute of Infectious Diseases (#1132, #1321). This study was conducted in accordance with the principles of the Declaration of Helsinki. All volunteers provided written informed consent prior to enrollment.

### Single B-cell sorting and culture
Single B cells were sorted and cultured as previously described with slight modifications[31]. Briefly, PBMCs were stained with BUV395-labeled anti-CD19 (clone HIB19), BV510-labeled anti-CD2 (clone RPA-2.10), CD4 (clone RPA-T4), anti-CD10 (clone HI10a), anti-CD14 (clone M5E2), anti-IgD (clone IA6-2), BV421-labeled anti-IgG (clone G18-145), live/dead aqua (Thermo Fisher Scientific), BB790-labeled anti-CD27 (clone O323), APC-labeled ancestral RBD, and PE-labeled Beta RBD at 1:200 dilution. All the antibodies were purchased from BD Biosciences and BioLegend. CD19+ CD2- CD4- CD10- CD14- IgD- CD27+ IgG+ Ancestral RBD+ Beta RBD+ B cells were single-cell sorted onto precultured MS40L-low feeder cells[34] in 96 F plates containing RPMI 1640 medium supplemented with 10% FBS, 55 μM 2-mercaptoethanol (2-ME), penicillin (100 U/mL), streptomycin (100 μg/mL), 10 mM HEPES, 1 mM sodium pyruvate, 1% minimal essential medium non-essential amino

acids, recombinant human interleukin-2 (IL-2; 50 ng/mL; PeproTech), recombinant human IL-4 (10 ng/mL; PeproTech), recombinant human IL-21 (10 ng/mL; PeproTech), recombinant human B-cell activating factor belonging to the TNF family (BAFF) (10 ng/mL; PeproTech) using FACS Symphony S6 (BD Biosciences). After culturing at 37 °C with 5% $CO_2$ for 24 days, the culture supernatant was collected and subjected to antibody characterization, and RNA was collected from the cells using RNeasy (Qiagen) for immunoglobulin gene sequencing.

## Immunoglobulin gene sequencing

Immunoglobulin gene sequencing was performed as previously described[33]. Briefly, RNA was extracted from each clone using the RNeasy micro kit (Qiagen) and reverse-transcribed using Super Script III (Thermo Fisher Scientific). Variable regions of the heavy and light chains were amplified by primer mix, and Sanger sequencing was performed by Azenta. Sequence files were analyzed using IgBlast (https://www.ncbi.nlm.nih.gov/igblast/).

## Enzyme-linked immunosorbent assay (ELISA)

Nunc MaxiSorp flat-bottom plates (Thermo Fisher Scientific) were coated with 1 µg/mL recombinant RBD at 4 °C overnight. After blocking with PBS containing 1% BSA and 0.05% Tween-20, serially diluted samples were applied to the plates and incubated for 2 h at room temperature. After washing, goat anti-human IgG-horseradish peroxidase (1:5000 dilution, HRP, Southern Biotech) diluted with Can Get Signal 2 (Toyobo) was added to the plates, and HRP activity was visualized with o-phenylenediamine dihydrochloride substrate (Sigma–Aldrich). After stopping the reaction with 2 N $H_2SO_4$, the optical density at 490 nm was measured using an Epoch2 spectrophotometer (Biotek). For single-cell culture supernatant ELISA, the average signal value of the media-only control multiplied by three was used as a threshold.

## Electrochemiluminescence immunoassay

Plasma antibody titers for mutant RBDs were measured using U-PLEX kits (Meso Scale Discovery), according to the manufacturer's instructions. Briefly, recombinant biotinylated RBDs were incubated with the linker proteins. After mixing with the stop solution, the linker-conjugated RBDs were added to U-PLEX plates, and the plates were incubated at 4 °C overnight. After washing with wash buffer, the plates were incubated with MSD Blocker A reagent (Meso Scale Discovery) to reduce non-specific binding, and diluted samples were added after washing. After incubation at room temperature for 2 h with rotation, the plates were incubated with SULFO-TAG-conjugated anti-human IgG (1:200 dilution, Meso Scale Discovery). After washing, MSD Gold read buffer B (Meso Scale Discovery) was added to the plates, and electrochemiluminescence was determined using MESOQuickPlex SQ 120 (Meso Scale Discovery).

## Neutralization assay (authentic virus, pseudovirus)

For authentic virus neutralization, the authentic virus (hCoV-19/Japan/TY-WK-521/2020, Wuhan strain), Delta variant (hCoV-19/Japan/TY11-927-P1/2021), BA.1 variant (hCoV-19/Japan/TY38-873P0/2021), BA.2 variant (hCoV-19/Japan/TY40-385-P1/2022), BA.5 variant (hCoV-19/Japan/TY41-702-P1/2022), BA.2.75 variant (hCoV-19/Japan/TY41-716-P1), BQ.1.1 variant (hCoV-19/Japan/TY41-796-P1/2022), XBB.1 variant (hCoV-19/Japan/TY41-795-P1/2022), and XBB.1.5 variant (hCoV-19/Japan/23-018-P1/2022) were mixed with diluted samples and added to VeroE6/TMPRSS2 cells (JCRB #1819) as previously described[10,13]. After culturing for 4–6 days, the cells were fixed with 20% formalin (Fujifilm Wako Pure Chemicals) and stained with a crystal violet solution (Sigma–Aldrich). The highest reciprocal dilution titer with >50% crystal violet staining was presented as the neutralization titer. $IC_{50}$ was examined with Cell Counting Kit-8 (Dojindo) for some experiments. After culturing for 2 days (3 days for BA.1, BA.5, XBB.1, and XBB.1.5),

Cell Counting Kit-8 was added to the culture, and the optical density at 450 nm was measured after culturing for one hour. $IC_{50}$ was calculated using Prism 9 (GraphPad). The experiment was performed with two to four replicates at the NIID BSL3 facility.

Pseudotyped reporter virus assays were conducted as previously described[31,35]. Using a plasmid encoding the SARS-CoV-2 spike protein (Addgene #145032) as a template, the D614G mutant, Omicron subvariants (BA.1, BA.2, and BA.5), and some candidate mutations from the DMS study were cloned into pcDNA4TO (Invitrogen) in the context of ΔC19 (19 amino acids deleted from the C-terminus)[36]. Spike protein-expressing pseudoviruses with a luciferase reporter gene were prepared by transfecting plasmids (pcDNA4TO Spike-ΔC19, psPAX2 (Addgene #12260), and pLenti firefly) into LentiX-293T cells using Lipofectamine 3000 (Invitrogen). After 48 h, the supernatants were harvested, filtered with a 0.45 µm low protein-binding filter (SFCA), and frozen at −80 °C. 293 T/ACE2 cells were seeded at a density of 10,000 cells/well in 96-well plates. Pseudoviruses and a three-fold dilution series of therapeutic agents were incubated for 1 h, and these mixtures were added to the 293 T/ACE2 cells. After 1 h of incubation, the medium was replaced. At 48 h post-infection, the cellular expression of the luciferase reporter, indicating viral infection, was determined using the ONE-Glo Luciferase Assay System (Promega). Luminescence was measured using an Infinite F200 Pro System (Tecan). Otherwise, vesicular stomatitis virus (VSV)-pseudovirus with SARS-CoV-2 variant spike protein was generated as previously described[37] by transfecting the pCAGGS SARS-CoV-2 spike expression vector into 293 T cells, followed by infection with G-complemented VSV ΔG/Luc. Culture supernatants containing VSV-pseudovirus were used for the neutralization assay. Pseudoviruses were mixed with antibody samples for 1 h and then added to VeroE6/TMPRSS2 cells. Luciferase activity in the cells was measured after 24 h of culture using the Bright-Glo luciferase assay system (Promega) and a GloMax Navigator Microplate Luminometer (Promega). This assay was performed in three to four replicates, and the nonlinear regression curve was calculated using Prism 9 (GraphPad).

## In vivo treatment

Syrian hamsters (4–5 weeks old females purchased from Japan SLC) were infected and treated as previously described[10]. Briefly, $10^4$ $TCID_{50}$ virus cells were intranasally inoculated into hamsters, and 5 mg/kg monoclonal antibody was intraperitoneally administered the following day. Ancestral virus-infected hamsters were monitored for body weight every day for 6 days. Nasal washes were collected from Omicron variant-infected hamsters 3 days after infection. RNA was isolated from the nasal wash using Direct-zol RNA miniprep (Zymo Research). Subgenomic viral RNA was quantified by qRT-PCR using the QuantiTect Probe RT-PCR Kit (Qiagen) with the following primers and the probe: sgLeadCoV2.Fwd 5'-CGATCTCTTGTAGATCTGTTCTC-3', E_Sarbeco_R 5'-ATATTGCAGCAGTACGCACACA-3', E_Sarbeco_P1 5'-VIC- AC ACTAGCCATCCTTACTGCGCTTCG -MGBNFQ-3'[18]. Infection experiments were performed at the NIID BSL3 SPF animal facility in accordance with the guidelines of the Institutional Animal Care and Use Committee of NIID.

## Biolayer interferometry

Kinetic assays were performed by capturing recombinant biotinylated RBDs on Octet SA biosensors (Sartorius), and association with recombinant Fabs (in-house) and carrier-free recombinant human ACE2 (BioLegend) was measured in 1× Octet kinetics buffer (Sartorius) using OCTET R8 (Sartorius).

## Cryo-electron microscopy (Cryo-EM) sample preparation and data collection

To prepare a complex sample for cryo-EM, the ancestral SARS-CoV-2 spike 6 P protein solution was incubated at 37 °C for 1 h before use. The

ancestral SARS-CoV-2 spike 6 P protein was expressed and purified using a previously reported procedure[10]. The purified Fab fragments of NIV-8, NIV-10, NIV-11, or NIV-13 were incubated with the ancestral SARS-CoV-2 Spike 6 P protein at a molar ratio of 1:3.2 at 18 °C for 1 h. A 0.1% (w/v) octyl-maltoside fluorinated solution (Jena Bioscience) was added to Spike-NIV-8, NIV-11, or NIV-13 solution to a final concentration of 0.01%, 0.03%, 0.01%, respectively. The sample was then applied to a Quantifoil R1.2/1.3 Cu 300 mesh grid (Quantifoil Micro Tools GmbH), which was freshly glow-discharged for 60 s or 90 s at 10 mA using a PIB-10 (vacuum device). The samples were plunged into liquid ethane using a Vitrobot Mark IV (Thermo Fisher Scientific) with the following settings: temperature, 18 °C; humidity, 100%; blotting time, 5 s; and blotting force, 5.

Micrographs were collected on a Krios G4 (Thermo Fisher Scientific) operated at 300 kV with a K3 direct electron detector (Gatan) at a nominal magnification of 130,000 (0.67 per physical pixel) using a GIF-Biocontinuum energy filter (Gatan) with a 20-eV slit width. Each micrograph was collected with a total exposure of 1.5 s and a total dose of 57.32 or 51.41 e/Å$^2$ over 50 frames. A total of 1986, 3110, 5170, and 3186 micrographs of Spike-NIV-8, Spike-NIV-10, Spike-NIV-11, and Spike-NIV-13 complexes, respectively, were collected at a 0° stage tilt. To address the preferred orientation in the dataset of the Spike-NIV-13 complex, a total of 1656 micrographs were additionally collected at a 40° stage tilt. All micrographs were collected at a nominal defocus range of 1.0–2.0 or 0.8–2.3 μm using EPU software (Thermo Fisher Scientific).

## Cryo-EM image processing

For the Spike-NIV-8 complex, micrograph movie frames were aligned, dose-weighted, and CTF-estimated using patch motion correction and patch CTF in CryoSPARC v3.3.1[38]. A total of 206,354 particles were blob-picked, and reference-free 2D classification ($K = 150$, batch = 300, iteration = 30) was performed to remove junk particles. Heterogeneous refinement was performed using the ab initio model, reconstructed with selected good particles after 2D classification in cryoSPARC. Non-uniform refinement was performed for the Spike-NIV-8 complex to align the particles, but the upper RBD region of the map was blurred due to flexibility. To classify RBD, a 3D classification focused on RBD without alignment was performed using cryoSPARC v3.1.1. Subsequently, the classes of down RBD and up RBD-bound NIV-8 were selected and processed by non-uniform refinement in cryoSPARC to generate the final cryo-EM maps. To improve the density of the Spike-NIV-8 interface, the aligned particles were symmetry-expanded, and a soft mask encompassing the down RBD and NIV-8 variable domains was made in UCSF Chimera[39]. The particles were imported to Relion 3.1[40] and subjected to focus 3D classification without alignment in Relion. Following a single round of focus 3D classification, particles belonging to the best class were imported back to cryoSPARC and subjected to local refinement, yielding a map with a global resolution of 3.4 Å.

For the Spike-NIV-10 complex, the dataset was preprocessed to 2D classification in the same manner as Spike-NIV-8, with 485,798 picked particles. Initial 3D models were reconstructed with ab initio reconstruction using particles belonging to 2D classes that showed a representative spike protein. All blob-picked particles were used for the first round of heterogeneous refinement using the four initial models in cryoSPARC. The Spike-NIV-10 map obtained showed a clear NIV-10 bound to RBD in the up conformation, but the other RBDs were unclear. Therefore, three rounds of heterogeneous refinement were performed to isolate spike proteins in different NIV-10 binding states. Each state was selected and processed by non-uniform refinement using cryoSPARC to generate the final cryo-EM maps. To improve the density of the Spike-NIV-10 interface, a soft mask encompassing RBD in the up-conformation and NIV-10 variable domains was created in the UCSF Chimera. After the first round of heterogeneous refinement, the

particles belonging to the Spike-NIV-10 class were selected and aligned with non-uniform refinement. These particles were imported into Relion 3.1, and the particles were subjected to focus 3D classification without alignments. Following a single round of focus 3D classification, particles belonging to the best class were imported back to cryoSPARC and subjected to local refinement, yielding a map with a global resolution of 4.2 Å.

For the Spike-NIV-11 complex, the dataset was preprocessed to 2D classification in the same manner as Spike-NIV-8, with 1,498,301 picked particles. Heterogeneous refinement was performed using the ab initio model reconstructed with selected good particles after 2D classification in cryoSPARC. Non-uniform refinement was performed for the Spike-NIV-11 complex to generate the final cryo-EM map with C3 symmetry. To improve the density of the Spike-NIV-11 interface, the aligned particles were symmetry-expanded, and a soft mask encompassing the up RBD and NIV-11 Fab region was made in UCSF Chimera. The particles were imported to Relion 3.1 and subjected to focus 3D classification without alignment. Following a single round of focus 3D classification, particles belonging to the best class were imported back to cryoSPARC and subjected to local refinement, yielding a map with a global resolution of 3.4 Å.

For the Spike-NIV-13 complex, the two datasets, collected at 0° and 40° stage tilt, were preprocessed separately for 2D classification, the same as in Spike-NIV-8 with 766,695 picked particles. The particles belonging to classes that show representative Spike proteins from 2D classifications with 0° and 40° datasets were selected and subsequently refined against a map of SARS-CoV-2 spike protein (EMDB: EMD-21452[41]). Three initial junk 3D models were reconstructed with ab initio reconstruction using particles from bad 2D classes. All blob-picked particles were used for the first round of heterogeneous refinement using spike protein reconstruction and three initial models in cryoSPARC. Particles from the Spike-NIV-13 class were subjected to another ab initio reconstruction to generate four reference maps. A Spike-NIV-13 map and four reference maps were generated for the first round of the heterogeneous refinement. Several rounds of heterogeneous refinement were performed to isolate spikes in different NIV-10 binding states. Each state was selected and processed by non-uniform refinement in cryoSPARC to generate the final cryo-EM maps. To improve the density of the Spike-NIV-13 interface, a soft mask encompassing the down RBD and NIV-10 variable domains was created in the UCSF Chimera. The same process as that for NIV-10 was used for local refinement to obtain the final map with a global resolution of 4.1 Å.

The reported resolutions are based on the gold-standard Fourier shell correlation curve (FSC = 0.143) criterion. The workflow of the data processing is shown in Supplementary Figs. 3–6. Data processing figures and final reconstructed maps were prepared using Chimera (version 1.15)[39] and Chimera X (version 1.1)[42].

## Cryo-EM model building and analysis

The structural models of the NIV-8 and NIV-11 Fab fragments were predicted using AlphaFold[43]. These predicted Fab models, SARS-CoV-2 Spike (PDB:6VYB[41], PDB:7K4N[25]), and SARS-CoV-2 Spike RBD domain (PDB:6M0J[44], PDB:7K45[25]) were fitted to the corresponding maps using UCSF Chimera. Iterative rounds of manual fitting in Coot (version 0.9.6)[45] and real-space refinement in Phenix (version 1.20)[46] were performed to improve the non-ideal rotamers, bond angles, and Ramachandran outliers. The final model was validated using MolProbity[47]. NIV-13 Fab fragment was modeled in the same manner as the NIV-8 or NIV-11 processes, and this model was only used to identify RBD epitope residues. The surface, cartoon, and stick representations of structural models shown in the figure's surface, cartoon, and stick representations were prepared using PyMOL (version 2.3.3) (http://pymol.sourceforge.net).

## X-ray crystal structure analysis

The deglycosylated RBD was mixed with NIV-10 at a molecular ratio of 1:1.3. The crystallization screening was performed using commercially available screening kits. Needle-like crystals were obtained in a condition containing 0.2 M sodium nitrate and 20% PEG3350. The X-ray diffraction experiment was performed using the Swiss Light Source beamline X06SA (Switzerland). The X-ray diffraction dataset was processed using XDS, followed by scaling using Aimless in the CCP4 package[48–50]. Molecular replacement was performed using Phaser in the PHENIX package while using the previously reported data on the structures of RBD and NIV-10 Fab prepared using alphafold2. The initial model was refined with rigid body refinement using "phenix.refine". The Fv region of the resultant model fitted the electron density well, whereas the CH1 and CL regions did not. Therefore, the CH1 and CL regions were deleted from the model and reconstructed using the "autobuild" function in the PHENIX package. Structural refinement was carried out using "phenix.refine" and COOT. The stereochemical properties of the structure were assessed using Mol-Probity. Figures were prepared with PyMOL (http://pymol. sourceforge.net).

## RBD deep mutational scanning for escape from monoclonal antibodies

Monoclonal antibody selection experiments were performed in duplicate using a deep mutational scanning approach with previously described mutant RBD libraries[20]. The library focused on the original Wuhan strain spike residues F329–C538, forming the RBD. Pooled oligos with degenerate NNK codons were synthesized by Integrated DNA Technologies Inc. The synthesized oligos were extended by overlap PCR and cloned into pcDNA4TO HMM38-HA-full-length spike plasmids. Transient transfection conditions that typically provided no more than a single coding variant per cell were used[51]. Expi293F cells at $2 \times 10^6$ cells per mL were transfected with a mixture of 1 ng of library plasmid and 1 μg of empty pMSCV plasmid per mL using ExpiFectamine (Thermo Fisher Scientific). Twenty-four hours after transfection, the cells were incubated with human ACE2 (hACE2)-harboring green fluorescent protein (GFP) reporter viruses, which were generated by transfecting pcDNA4TO hACE2, psPAX2 (Addgene #12260), and pLenti GFP into LentiX-293T cells using Lipofectamine 3000 (Invitrogen). Viruses carrying hACE2 instead of glycoproteins can infect cells expressing the spike protein. To analyze the escape mutations from antibodies, cells were pre-incubated with neutralizing antibodies for 1 h. Next, these cells were treated with hACE2-harboring viruses for 1 h and further incubated with fresh medium for 24 h. Cells were harvested, washed twice with PBS containing 10% bovine serum albumin (BSA), and then co-stained for 20 min with anti-hemagglutinin (HA) Alexa Fluor 647 (clone TANA2, MBL). The cells were washed twice before sorting on an MA900 cell sorter (Sony). Dead cells, doublets, and debris were excluded by first gating the main population using forward and side scatter. GFP-positive and GFP-negative cells were collected from the HA-positive (Alexa Fluor 647 positive) population. The total number of collected cells was approximately 2 million per group. Total RNA was extracted from the collected cells using TRIzol (Life Technologies) and Direct-zol RNA MiniPrep (Zymo Research Corporation), according to the manufacturer's protocol. First-strand complementary DNA (cDNA) was synthesized using PrimeScript II Reverse Transcriptase (Takara) primed with a gene-specific oligonucleotide. Libraries were separately designed for the three sections in the RBD and then pooled to analyze the same experimental conditions. After the cDNA synthesis, each library was amplified using specific primers. Following a second round of PCR, adapters for annealing were added to the Illumina flow cell, together with barcodes for each sample identification. The PCR products were sequenced on an Illumina NovaSeq 6000 using a 2 × 150 nucleotide paired-end protocol at the Department of Infection Metagenomics, Research Institute for Microbial Diseases, Osaka University. Data were analyzed by comparing the read counts of each group normalized relative to the wild-type sequence read count. $\log_{10}$ enrichment ratios for all individual mutations were calculated and normalized by subtracting the $\log_{10}$ enrichment ratio for the wild-type sequence across the same PCR-amplified fragment.

## Escape virus selection and limited-dilution cloning

Escape virus selection was performed as previously described[10]. Briefly, the authentic virus or BA.1 virus was passaged on VeroE6/TMPRSS2 cells in the presence of neutralizing antibodies. Then, the culture supernatant was collected from cells with CPE at the highest antibody concentration. The culture supernatant containing escape viruses was diluted $10-10^8$ times and added to VeroE6/TMPRSS2 supplemented with selection antibody in 96-well flat-bottom plates. After 4–5 days of culture, the culture supernatants containing the cloned virus were collected from plates with <10 CPE wells. Viral RNA was extracted using Direct-zol RNA miniprep (Zymo Research), Sanger-sequenced after reverse transcription with LunaScript RT SuperMix (New England Biolabs), and amplified with PrimeSTAR Max DNA Polymerase (Takara Bio). The forward primer (nCoV-2019_74_LEFT, 5′-ACATCACTAGGTTTCAAACTTTACTTGC-3′) and reverse primer (nCoV-2019_77_RIGHT, 5′-CAGCCCCTATTAAACAGCC TGC-3′) were obtained from the ARTIC network (https://artic.network/ncov-2019), and sequence data were analyzed for mutations using CoVsurver on GISAID (https://www.gisaid.org/epiflu-applications/covsurver-mutations-app/). For Next-Generation Sequencing (NGS), the whole genome of SARS-CoV-2 was amplified using a modified version of the ARTIC Network protocol for CoV-2 genome sequencing[52]. An NGS library was constructed and sequenced using the QIAseq FX DNA Library Kit (Qiagen) and MiSeq System (Illumina). Consensus sequences of the viral genome were obtained using the ARTIC field bioinformatics pipeline, following the ARTIC-nCoV-bioinformatics SOP-v1.1.0 (https://artic.network/ncov-2019/ncov2019-bioinformatics-sop.html, 2020).

## Molecular dynamics (MD) simulations and hydrophobicity calculation

MD simulations of the RBD in complex with each antibody (NIV-10 and −13), as well as the RBD alone, were performed using GROMACS 2018.6[53] with the CHARMM36m force field[54]. The initial structures of the NIV-10 and −13 complexes were obtained from the cryo-EM structures. Each system was solvated with TIP3P water in a rectangular box, so the minimum distance to the edge of the box was 10 Å under periodic boundary conditions. Na and Cl ions were added to neutralize the protein charge; next, additional ions were added to mimic a salt solution concentration of 0.15 M. Each system was energy-minimized for 5000 steps using steepest descent, heated from 50 to 310 K for 200 ps, and further equilibrations were continued for 500 ps with the NVT ensemble. During equilibration, positional restraint potentials were applied, and their force constants were gradually reduced. Further production runs were performed using the NPT ensemble. A cutoff distance of 12 Å was used for Coulomb and van der Waals interactions. The long-range electrostatics were evaluated using the particle mesh Ewald method[55]. The LINCS algorithm was employed to constrain hydrogen atoms bonds[56]. The time step was set as 2 fs throughout the simulation. A simulation was repeated six and three times for the antibody-RBD complex (200 ns each) and RBD (100 ns each) systems, respectively, which resulted in approximately 3 μs of aggregate simulation data, and the snapshots were saved every 100 ps. During the last 50 ns trajectories, the standard deviations of the root mean square deviation (RMSD) for the Cα atoms of the antibodies and RBD were within 1.0 Å (Supplementary Fig. 11). Therefore, to allow relaxation from the starting structures, all trajectory analyses were performed based on the last 50 ns trajectories, through the Gromacs,

Prody[57], and MDTraj packages[58]. The UCSF Chimera[39] was used to visualize the MD trajectories.

The hydrophobicity of the antigen-binding site of each antibody and RBD was assessed using the spatial aggregation propensity (SAP) score[59] with the CHARMM program[60], starting either from the cryo-EM structures (antibodies: NIV-13[EMD-33830], Omi-2 [7ZR9], Omi-42 [7ZR7]), X-ray crystal structures (antibodies: NIV-10[8HES], NIV-11[8HGL], Omi-3 [7ZF3], Omi-18 [7ZFB], Omi-25 [7ZFD], BD-515 [7E88]; RBDs: ancestral [8HES], BA.5 [7ZXU], BQ.1.1 [7ZXU], XBB [7ZF7], XBB.1.5 [7ZF7]) or predicted model structures (the other NIV antibodies) generated with DeepAb[61]. Based on the aforementioned crystal structures as templates, the side chains of BQ.1.1, XBB, and XBB.1.5 were modeled using the rotamer library in the Rosetta software suite[62] To account for the variety of resolutions of the initial structures, the SAP radius was set to 10 Å, which can identify low-resolution broader hydrophobic patches.

### Docking simulations

Docking simulations between NIV-10 and RBD variants were performed with the Rosetta software suit[62]. Only high-resolution refinement was employed. Initial backbone coordinates were obtained from the crystal structure of the NIV-10/Ancestral RBD complex [8HES]. The side chains of XBB and XBB.1.5 was modeled using the rotamer library in Rosetta.

### Computer-aided antibody design

The NIV-10 antibody sequence design was carried out using the Rosetta software suite[62]. Initial coordinates for the design calculations were derived from the crystal structure of the NIV-10 Fab-RBD complex, with only the Fv region of the antibody and RBD used as input. As the primary objective was to enable NIV-10 to bind to XBB.1.5, the ancestral RBD sequence was substituted with that of XBB.1.5, resulting in a tentative NIV-10-XBB.1.5 RBD complex model. This complex model was subsequently refined using the FastRelax protocol with all-atom constraints[63] followed by a sequence design step where CDR residues within 5 Å of the RBD were permitted to change using the FastDesign protocol with the InterfaceDesign2019 script[64].

The 2000 generated complex models were then subjected to docking calculations[65] to assess the complementarity at the interface between the designed NIV-10 and a series of RBD variants, such as XBB.1.5, XBB, and ancestral. Selection criteria for the interfaces included interface energy (<−35 kcal/mol), shape complementarity (Sc > 0.65), and the number of hydrogen bonds across the interfaces (>4)[66]. After thorough visual examination of the designed antibody-RBD complex structures, an antibody meeting these criteria for all RBD variants was selected for experimental validation.

### Statistical analysis

All statistical analysis was performed with Prism 9 (Graphpad) unless specified otherwise.

### Reporting summary

Further information on research design is available in the Nature Portfolio Reporting Summary linked to this article.

## Data availability

The DMS data generated in this study have been deposited at SpikeDB (https://sysimm.ifrec.osaka-u.ac.jp/sarscov2_dms/) and the National Center for Biotechnology Information Sequence Read Archive under BioProject ID PRJNA970973. Atomic coordinates and cryo-EM maps of the reported structure have been deposited into the Protein Data Bank and Electron Microscopy Data Bank (Supplementary Table 6). For SARS-CoV-2 spike in complex with NIV-8, states 1, 2, and RBD-Fab (local refinement) were assigned as EMD-33821, EMD-33822 (PDB 7YH7), EMD-33820 (PDB 7YH6), respectively. For SARS-CoV-2 spike in

complex with NIV-10, states 1, 2, 3, and RBD-Fab (local refinement) were assigned as EMD-33824, EMD-33825, EMD-33826, EMD-33823, respectively. For SARS-CoV-2 spike in complex with NIV-10, the crystal structure of the RBD complex was assigned as PDB 8HES. For SARS-CoV-2 spike in complex with NIV-11, 3-up state and RBD-Fab (local refinement) were assigned as EMD-34741 (PDB 8HGL), EMD-34732 (PDB 8HGM), respectively. For SARS-CoV-2 spike in complex with NIV-13, states 1, 2, 3, and RBD-Fab (local refinement) were assigned as EMD-33828, EMD-33829, EMD-33830, respectively. All other data are available in the main text or the supplementary materials. Source data are provided with this paper.

## Code availability

The MD trajectories and the model structure of the FD03/XBB.1.5 RBD complex, along with the sequence information, have been submitted to the Biological Structure Model Archive (BSM-Arc) under BSM-ID BSM000046 [https://bsma.pdbj.org/entry/46][67].

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

## Acknowledgements

We thank E. Izumiyama, A. Dosaka, K. Isoyama, N. Yoshida for technical support, and R. Itami for administrative assistance. We also acknowledge the Paul Scherrer Institut, Villigen, Switzerland for the provision of synchrotron radiation beamtime at beamline X06SA of the SLS and would like to thank Takashi Tomizaki for assistance. The supercomputing resources in this study were provided in part by the Human Genome Center at the Institute of Medical Science, The University of Tokyo, and by Research Center for Computational Science, Okazaki, Japan (Project: 22-IMS-C091). This work was supported by Japan Agency for Medical Research and Development grants JP20fk0108516 (to SM, T.H., and K. Maenaka), JP21fk0108465 (to A.H.), JP20fk0108298 (to T.H., K. Maenaka, and Y.T.), JP20fk0108534 (to K. Maenaka and Y.T.), JP21fk0108534 (to K. Maenaka and Y.T.), JP19fk0108104 (to Y.T.), JP20fk0108104 (to Y.T.), JP22gm1810004 (to K. Maenaka and Y.T.), JP22fk0108141 (to Y.T.), JP22ama121037 (to K. Maenaka), JP223fa627005 (to K. Maenaka), JP22wm0325047 (to D.K.), JP223fa627009 (to T.H.); Japan Society for the Promotion of Science KAKENHI grant number JP20H05873 (to K. Maenaka), JP20H05773 (to T.H.), JP19H04202 (to D.K.); Takeda Science Foundation (to K. Maenaka); and Joint Usage/Research Center program of Institute for Life and Medical Sciences, Kyoto University (to K. Maenaka); Japan Science and Technology Agency Core Research for Evolutionary Science and Technology grant number JPMJCR20H8 (to T.H. and H.F.).

## Author contributions

S.M., Y. Anraku, S.T., Y. Adachi, D.K., Y.H., Y. Kirita, R.K., K.T., K.Y., T. Suzuki, S.K., T. Someya, H.F., Y. Kuroda, T.Y., T.O., S.F., K. Maeda, T.H., A.H. performed research, collected, and analyzed data; F.N.-U. provided human samples; S.M., T.H., K. Maenaka, Y.T. conceived, designed, and coordinated the study; S.M., Y. Anraku, D.K., S.F., A.H., K. Maenaka, Y.T. wrote and edited the manuscript.

## Competing interests

S.M., Y.A., K.Y., D.K., and Y.T. declare that an intellectual property application has been filed using the data presented in this paper. Other authors declare that they have no competing interests.

## Additional information

[1]Research Center for Drug and Vaccine Development, National Institute of Infectious Diseases; Shinjuku-ku, Tokyo 162-8640, Japan. [2]Laboratory of Biomolecular Science, and Center for Research and Education on Drug Discovery, Faculty of Pharmaceutical Sciences, Hokkaido University; Sapporo, Hokkaido 060-0812, Japan. [3]Department of Cardiovascular Medicine, Graduate School of Medical Science, Kyoto Prefectural University of Medicine; Kyoto, Kyoto 602-8566, Japan. [4]Department of Nephrology, Graduate School of Medical Science, Kyoto Prefectural University of Medicine; Kyoto, Kyoto 602-8566, Japan. [5]Department of Life Science and Medical Bioscience, Waseda University; Shinjuku-ku, Tokyo 162-8480, Japan. [6]Laboratory of Medical Virology, Institute for Life and Medical Sciences, Kyoto University; Kyoto, Kyoto 606-8507, Japan. [7]Division of Pathogen Structure, International Institute for Zoonosis Control, Hokkaido University, Sapporo 001-0020, Japan. [8]Department of Veterinary Science, National Institute of Infectious Diseases; Shinjuku-ku, Tokyo 162-8640, Japan. [9]Department of Virology I, National Institute of Infectious Diseases; Shinjuku-ku, Tokyo 162-8640, Japan. [10]Department of Infectious Diseases, Tokyo Metropolitan Bokutoh Hospital; Sumida-ku, Tokyo 130-8575, Japan. [11]Global Station for Biosurfaces and Drug Discovery, Hokkaido University; Sapporo, Hokkaido 060-0812, Japan. [12]Institute for Vaccine Research and Development (HU-IVReD), Hokkaido University; Sapporo, Hokkaido 060-0812, Japan. ✉e-mail: sayamrym@niid.go.jp; ytakahas@niid.go.jp

