## [Peer review file · Nature Communications]

REVIEWER COMMENTS

Reviewer #1 (Remarks to the Author):

Moriyama et al. manuscript describes isolation of antibodies from convalescent individuals with evolutionary constrained properties, namely, antibodies that bind ACE2 binding site in the RBD and have low mutation frequency in circulating variants. From 16 isolated broadly neutralizing antibodies they determine that antibodies targeting Y489 but not binding to F486 retain the broadest neutralization potency. Finally, they show that hydrophobic H-CDR3 in NIV-10 antibody (the broadest neutralizing antibody in the isolated panel), may be responsible for facilitating interaction with the hydrophobic part of RBD and limiting virus escape. The manuscript is well written and experiments support the conclusions. These I my suggestions for improving the manuscript:

1. Some statements in the 1st paragraph of the introduction need citations.
2. Line 11 on page 3 authors use incorrect acronym: RBD is short for receptor binding domain, RBS is short for receptor binding site.
3. In this statement -Furthermore, sites in the RBS that are scarcely mutated in circulating viruses in the real world are more likely to be vulnerable to viruses - authors probably mean vulnerable to antibodies not viruses.
4. Figure 3a escape logo plots might be better colored by ACE2 binding or natural mutation frequencies as opposed to chemical amino acid properties given that authors are looking for antibodies that would bind evolutionary vulnerable sites.
5. On page 6 line 2 authors say that NIV-10 escape was seen exclusively at site G485P. While yes G485P completely escapes NIV-10 it is too strong of a statement that the escape is exclusive to this site as the authors show in fig. 3d other mutations have almost 10-fold increase in IC50 as well.
6. Authors find no escape against NIV-11 in live virus assay, it would be good to validate the DMS profiles for this antibody at least with a pseudovirus neutralization assay.

It would be useful for authors to add neutralization assay against 7. XBB.1.5 variant to figure 4a. This variant is the most up to date circulating strains and, importantly, it has F486P mutations in the epitope of NIV-11 and NIV-13 and antibodies and may have an effect on NIV-10 as well.

Reviewer #2 (Remarks to the Author):

The work by S. Moriyama and co-workers addresses the identification of antibodies able to neutralise SARS-CoV-2 variants, including Omicron BA.4/BA.5. In particular, the authors reported Y489 as a virus vulnerability spot and provided a stack of structural data of antibody-RBD complexes. The findings of the work are relevant, the methodologies are appropriate and described with enough details for reproducibility. The overall discussion and the results presentation are, however, a bit difficult to follow. I would support manuscript publication provided some revisions.

Below some major points:

The acronyms RBS and RBD are used within the manuscript to indicate the receptor binding domain; i would recommend to clarify such nomenclature within the manuscript and adding a figure to show the precise location (perhaps in addition to figure 1b), in order to help reader's understanding.

On page 8, lines 29-31 the authors state: "The F486 recognition modes of NIV-10 and NIV-11 were structurally different, and NIV-10 had more space around F486 than NIV-11, possibly accounting for the NIV-10 tolerability 31 to F486 mutations (Fig. 5f).

Could the authors better quantify such more space around F486?

I would recommend to include as supplementary figures the comparisons of RMSD from the different sets of simulations, also to better justify the choice to analyse the last 50 ns of trajectory.

I find Supplementary figure 8 very difficult to interpret, i would recommend the authors to re-organize the figure, maybe using a different representation and/or adding further labels.

On page 21, lines 10-11 the authors report: NIV-13, Omi-2 [7ZR9] , Omi-42 [7ZR7]), X-ray crystal structures (NIV-10, NIV-

11 11, Omi-3 [7ZF3], Omi-18 [7ZFB], Omi-25 [7ZFD] , BD-515 [7E88]).

I would recommend to use a consistent nomenclature for virus variants thorough the manuscript.

Since this journal is committed to sharing data associated with articles, upon paper acceptance, authors are encouraged to submit their MD data to a publicly available archive, such as BSM-Arc (<https://bsma.pdbj.org/>).

Minor points:

there are some typos, e.g on page 8, line 14 public chronotypes

pag 18, line 11 "and this model were only used"

pag 21, line 10 "from the cry-EM"

Reviewer #3 (Remarks to the Author):

The study by Moriyama and colleagues outlines the isolation of human monoclonal antibodies based on recognition of eight RBD mutations, including F486S, which are considered important ACE2 binding sites. The extensive analysis of more than 900 mAbs revealed that Y489 and F486 in the RBD were crucial contact residues for those with broad recognition of pre-Omicron and Omicron variants, and many of them were derived from "public" VH clonotypes (such as VH-1-58). Sixteen of the mAbs were divided into three groups based on recognition of three specific RBD mutations (Y489S, F486S, and S443N) and were further evaluated for neutralization breadth, escapability, epitope specificity, and efficacy in hamsters. One mAb, NIV-10, stood out due to its unique features, such as lack of escape from authentic virus, vulnerability limited to the mutation G485P, and recognition of a highly hydrophobic epitope in the "neck to shoulder" region of the RBD. Overall, NIV-10 could be a potential candidate for developing as a next-generation COVID-19 therapeutic.

The authors have employed a diverse range of experimental methods, including innovative ones, to carry out an extensive analysis of the isolated mAbs. These methods include a focused screening strategy, a novel inverted infection DMS assay to investigate both mAbs binding and viral fitness, and the application of molecular dynamics to evaluate the individual impact of epitope contact residues (such as F486).

A significant limitation of the study is the use of outdated variants to assess the binding and neutralizing properties of the most noteworthy mAbs, particularly NIV-10. This study does not consider the Omicron variants BA.2.75.2, BQ.1.1, and XBB.1.5, which is crucial for verifying that NIV-10 is resistant to F486 substitutions in currently prevalent variants, such as F486S in BA.2.75.2 and F486P in XBB.1.5. Another concern is that the escapability was evaluated using the ancestral virus to identify escapes, potentially overlooking the influence of epistatic effects if using an Omicron variant as the backbone.

Major comments:

- Title and abstract: The title and abstract of the study do not adequately emphasize the significance of the most important discovery, which is the distinctiveness of the NIV-10 antibody.
- Fig. 1a: consider revising the data in Panel A to reflect the evolution of the virus since April 2022.
- Fig. 1c: The legend states that 823 clones were analyzed, however, it appears that 947 antibodies were actually evaluated based on the Results section (line 5).
- Fig. 2b: For improved readability, it would be advisable to substitute the heatmap plot with a table or a graph that clearly presents the actual IC50 values.
- Fig. 2c-f: The study lacks clear information on the dosing regimen, route, and timing of antibody administration in the in vivo challenge study, please make the necessary adjustments.
- Fig. 3a: The effect of RBD mutations on the fitness of the virus is not clear in the figure presented. The statement that the NIV-10 escapes are solely caused by the mutation from G to P at site 485 is not entirely accurate, as other substitutions such as L455W/C or E484K also had an impact on binding, although to a lesser degree. It would be beneficial to determine which of these substitutions can occur as a result of a single nucleotide mutation.
- Fig 3b: The figure is difficult to read due to the small font size and legends, it is recommended to redo this figure for improved legibility.
- Fig. 3c: the top five NIV-10 escapes tested (D405M, Y449P/K, L455C and G485P) are not all listed in Fig. 3a
- Fig. 3d: as mentioned above the potential impact of epistatic effects, particularly with Omicron variants carrying mutations at position 486, on the effect of escape mutants and neutralization susceptibility should be taken into account. It may be beneficial to test these mutants on the Omicron backbone (against which the affinity of NIV-10 is reduced by more than 3-logs, as shown in Supplementary Table 3) or include a cautionary note in the interpretation of these results in the text.
- Page 10, lines 27-28: The statement that "These data suggest that current mRNA vaccination cannot efficiently boost or maintain Y489S- B cells" may be considered too speculative. It may be appropriate to tone it down or remove it.

Minor comments:

- Page 4, lines 1-2: Please provide an explanation for the choice of mutations and whether they represent the mutations with the greatest reduction in ACE2 binding.
- Page 4, line 2-3. The sentence "Five amino acids were positioned in the RBS, suggesting that they are critical for ACE2 binding" appears to be tautological in nature.

- Page 4, line 14: Please revise the text to accurately reflect the evolution of Omicron variants. BA.4/5 are no longer considered recent variants as they have been present for over six months and have been largely replaced by newer variants with different F486 mutations, such as XBB.1 (F486S), XBB.1.5 (F486P), BA.4.6 (F486S), and BA.2.75.2 (F486S).
- Page 6, line 24: please indicate that the mutation found in BA.4 and BA.5 is F486V
- Page 6, line 31: remove "the" in "the neutralizing activity"
- Page 6, line 6: this is consistent with epistatic effects of other Omicron mutations, as shown in the paper by Bloom and colleagues (Starr et al, doi.org/10.1371/journal.ppat.1010951). Please consider adding this quote.
- Page 7, lines 7-8: this statement is further supported by the recent data showing that the mutation of S486 in XBB.1 into P486 in XBB.1.5 resulted in a markedly increased affinity for huACE2, as reported in Yue et al. (10.1101/2023.01.03.522427).
- Page 8, line 8-9: consider replacing "showed" with "recognized"
- Page 10, line 10: The expression "the neck to the left shoulder" can be defined more clearly by replacing it with "the neck to the left shoulder RBS region."
- Page 10, lines 15-17: The sentence regarding the intriguing outcomes obtained from MD simulations may need to be better articulated and expanded for clarity.
- Page 10, line 2: this may be true for infections not just vaccinations.
- Fig. 3d: The plot is difficult to interpret due to indistinct colors, please consider improving it for better readability. The same for Figure 4 a-e plots.
- Fig. 6d: consider highlighting NIV-10 mAb in the plot

Response to reviewer #1:

Moriyama et al. manuscript describes isolation of antibodies from convalescent individuals with evolutionary constrained properties, namely, antibodies that bind ACE2 binding site in the RBD and have low mutation frequency in circulating variants. From 16 isolated broadly neutralizing antibodies they determine that antibodies targeting Y489 but not binding to F486 retain the broadest neutralization potency. Finally, they show that hydrophobic H-CDR3 in NIV-10 antibody (the broadest neutralizing antibody in the isolated panel), may be responsible for facilitating interaction with the hydrophobic part of RBD and limiting virus escape. The manuscript is well written and experiments support the conclusions.

We appreciate these positive comments from the reviewer.

1. Some statements in the 1st paragraph of the introduction need citations.

We have added citations (page 2, lines 24 and 27).

2. Line 11 on page 3 authors use incorrect acronym: RBD is short for receptor binding domain, RBS is short for receptor binding site.

We apologize for the confusion. We have modified the sentence on page 3, lines 12-14.

3. In this statement -Furthermore, sites in the RBS that are scarcely mutated in circulating viruses in the real world are more likely to be vulnerable to viruses - authors probably mean vulnerable to antibodies not viruses.

We apologize for the confusion. Please see modified sentence on page 3, lines 14-15.

4. Figure 3a escape logo plots might be better colored by ACE2 binding or natural mutation frequencies as opposed to chemical amino acid properties given that authors are looking for antibodies that would bind evolutionary vulnerable sites.

Thank you for the suggestion. We have changed the colors of the logo plots by natural mutation frequencies (Figure 3a; page 30, lines 5 to page 31, line 1).

5. On page 6 line 2 authors say that NIV-10 escape was seen exclusively at site G485P. While yes

G485P completely escapes NIV-10 it is too strong of a statement that the escape is exclusive to this site as the authors show in fig. 3d other mutations have almost 10-fold increase in IC50 as well.

Thank you for the comment. We agree with this comment and modified the sentence accordingly (page 6, lines 12-13).

6. Authors find no escape against NIV-11 in live virus assay, it would be good to validate the DMS profiles for this antibody at least with a pseudovirus neutralization assay.

Thank you for the comment. The DMS profile showed that F486 mutation was resistant to NIV-11, and indeed, NIV-11 poorly neutralized BA.5, which has an F486S mutation (Figure 4a, 4b).

It would be useful for authors to add neutralization assay against 7. XBB.1.5 variant to figure 4a. This variant is the most up to date circulating strains and, importantly, it has F486P mutations in the epitope of NIV-11 and NIV-13 and antibodies and may have an effect on NIV-10 as well.

Thank you for the valuable suggestion. We agree that the neutralization information for the most recent variants is relevant. When we calculated the hydrophobicity of recent variants, XBB RBS is less hydrophobic compared to other Omicron sublineages such as BA.5 and BQ.1.1, and XBB.1.5 RBS is slightly more hydrophobic than XBB due to proline mutation at a position 486 (newly added Fig. S9). We have performed neutralization assay of NIV-10 against BQ.1.1, XBB, and XBB.1.5, and found that NIV-10 potently neutralized BQ.1.1, but XBB mildly escaped NIV-10, which is consistent to less hydrophobicity of XBB RBS (newly added Fig. 7a and 7b). To our surprise (because XBB.1.5 RBS is more hydrophobic than XBB RBS), XBB.1.5 escaped NIV-10 even at a concentration of 100 µg/mL, which provoked us to investigate the correlation between the hydrophobicity of RBD and antibody binding.

Since XBB.1.5 has a new substitution at position 486, we made recombinant XBB RBD with a single amino acid substitution at a position 486 that can occur by a single nucleotide mutation from F486 and S486, and examined their antibody binding. XBB RBD with I486, L486, Y486, V486, A486, P486, T486, R486 mutation were examined for antibody binding by ECLIA. We also calculated the hydrophobicity of the mutant RBDs and confirmed that the SAP value and binding signal of each XBB RBD mutant to the antibodies were well correlated, except for the F486P mutant (XBB.1.5) to NIV-10 (newly added Fig. 7c). Structural modeling suggested the fitness of NIV-10 to XBB.1.5 was impeded by the proline mutation at position 486, mainly by a steric clash between P486 and the

antibody (newly added Fig. S10a).

To address this limitation of NIV-10, we computationally designed a new antibody sequence based on NIV-10, emphasizing interaction energy, shape complementarity, and the number of hydrogen bonds at the antibody-antigen interface. This resulted in the creation of a NIV-10-based designed antibody, NIV-10/FD03. Recombinant NIV-10/FD03 antibody potently bound to XBB RBDs regardless of mutations at position 486, potently bound to other previous VOCs, and neutralized XBB and XBB.1.5 (newly added Fig. 7d, 7e, and 7f). These new data are now shown in Figure 7, Fig. S9, Fig. S10 and described on page 10 line 8 to page 11 line 13, page 12 line 27 to page 13 line 9.

In addition, to make the computationally designed antibody public, the sequence and theoretical model structure of the designed antibody in complex with XBB.1.5 RBD have been submitted to the Biological Structure Model Archive (BSM-Arc) under BSM-ID BSM00046 (<https://bsma.pdbj.org/entry/46>).

We greatly appreciate the helpful comments of the reviewer; they have improved the manuscript substantially. We sincerely hope that this revised revision meets the criteria for publication in the Nature Communications.

Response to reviewer #2:

The work by S. Moriyama and co-workers addresses the identification of antibodies able to neutralise SARS-CoV-2 variants, including Omicron BA.4/BA.5. In particular, the authors reported Y489 as a virus vulnerability spot and provided a stack of structural data of antibody-RBD complexes. The findings of the work are relevant, the methodologies are appropriate and described with enough details for reproducibility. The overall discussion and the results presentation are, however, a bit difficult to follow. I would support manuscript publication provided some revisions.

Thank you for these positive comments on our manuscript.

- 1. The acronyms RBS and RBD are used within the manuscript to indicate the receptor binding domain; i would recommend to clarify such nomenclature within the manuscript and adding a figure to show the precise location (perhaps in addition to figure 1b), in order to help reader's understanding.*

We appreciate this suggestion and highlighted RBS in Figure 1b.

- 2. On page 8, lines 29-31 the authors state: "The F486 recognition modes of NIV-10 and NIV-11 were structurally different, and NIV-10 had more space around F486 than NIV-11, possibly accounting for the NIV-10 tolerability 31 to F486 mutations (Fig. 5f). Could the authors better quantify such more space around F486?"*

Thank you for the comment. We have quantified the space around F486 in NIV-10 and NIV-11 by calculating solvent-accessible surface area of F486. We found that NIV-10 has a larger value compared to NIV-11, indicating that F486 has more solvent accessible apace when interacting with NIV-10. The data is now shown in Fig. 5f and described in the manuscript (page 9 lines 12-14).

- 3. I would recommend to include as supplementary figures the comparisons of RMSD from the different sets of simulations, also to better justify the choice to analyse the last 50 ns of trajectory.*

Thank you for the suggestion. We have computed RMSD of C-alpha atoms for each trajectory to better justify our decision to analyze the last 50 ns of the trajectory. The results are detailed in the method section (page 23 lines 29-31) and can also be viewed in Figure S11.

- 4. I find Supplementary figure 8 very difficult to interpret, i would recommend the authors to re-*

organize the figure, maybe using a different representation and/or adding further labels.

Following the reviewer's comment, we have reorganized Fig S8 and have modified the figure legend.

5. *On page 21, lines 10-11 the authors report: NIV-13, Omi-2 [7ZR9] , Omi-42 [7ZR7]), X-ray crystal structures Page 5 of 6 (NIV-10, NIV-11 11, Omi-3 [7ZF3], Omi-18 [7ZFB], Omi-25 [7ZFD] , BD-515 [7E88]). I would recommend to use a consistent nomenclature for virus variants thorough the manuscript.*

Thank you for pointing this out. We have modified the description to clarify they are antibody name, and added the information of RBD which we missed in the first submission (page 24 lines 3-6).

6. *Since this journal is committed to sharing data associated with articles, upon paper acceptance, authors are encouraged to submit their MD data to a publicly available archive, such as BSM-Arc (<https://bsma.pdbj.org/>).*

The MD trajectories, along with the sequence and structure of the designed antibody, have been submitted to the Biological Structure Model Archive (BSM-Arc) under BSM-ID BSM000046 (<https://bsma.pdbj.org/entry/46>).

Minor points:

there are some typos, e.g on page 8, line 14 public chronotypes

pag 18, line 11 "and this model were only used"

pag 21, line 10 "from the cry-EM"

Thank you for pointing them out. We have corrected these typos.

We greatly appreciate the helpful comments of the reviewer; they have improved the manuscript substantially. We sincerely hope that this revised revision meets the criteria for publication in the Nature Communications.

Response to reviewer #3:

The study by Moriyama and colleagues outlines the isolation of human monoclonal antibodies based on recognition of eight RBD mutations, including F486S, which are considered important ACE2 binding sites. The extensive analysis of more than 900 mAbs revealed that Y489 and F486 in the RBD were crucial contact residues for those with broad recognition of pre-Omicron and Omicron variants, and many of them were derived from "public" VH clonotypes (such as VH-1-58). Sixteen of the mAbs were divided into three groups based on recognition of three specific RBD mutations (Y489S, F486S, and S443N) and were further evaluated for neutralization breadth, escapability, epitope specificity, and efficacy in hamsters. One mAb, NIV-10, stood out due to its unique features, such as lack of escape from authentic virus, vulnerability limited to the mutation G485P, and recognition of a highly hydrophobic epitope in the "neck to shoulder" region of the RBD. Overall, NIV-10 could be a potential candidate for developing as a next-generation COVID-19 therapeutic. The authors have employed a diverse range of experimental methods, including innovative ones, to carry out an extensive analysis of the isolated mAbs. These methods include a focused screening strategy, a novel inverted infection DMS assay to investigate both mAbs binding and viral fitness, and the application of molecular dynamics to evaluate the individual impact of epitope contact residues (such as F486).

We appreciate reviewer's constructive comments based on thorough reading on our manuscript. We hope we could have addressed all points raised by the reviewer accordingly.

- 1. A significant limitation of the study is the use of outdated variants to assess the binding and neutralizing properties of the most noteworthy mAbs, particularly NIV-10. This study does not consider the Omicron variants BA.2.75.2, BQ.1.1, and XBB.1.5, which is crucial for verifying that NIV-10 is resistant to F486 substitutions in currently prevalent variants, such as F486S in BA.2.75.2 and F486P in XBB.1.5.*

Thank you for the valuable suggestion. We agree that the neutralization information for the most recent variants is relevant. When we calculated the hydrophobicity of recent variants, XBB RBS is less hydrophobic compared to other Omicron sublineages such as BA.5 and BQ.1.1, and XBB.1.5 RBS is slightly more hydrophobic than XBB due to proline mutation at a position 486 (newly added Fig. S9). We have performed neutralization assay of NIV-10 against BQ.1.1, XBB, and XBB.1.5, and found that NIV-10 potently neutralized BQ.1.1, but XBB mildly escaped NIV-10, which is consistent to less hydrophobicity of XBB RBS (newly added Fig. 7a and 7b). To our surprise (because XBB.1.5 RBS is more hydrophobic than XBB RBS), XBB.1.5 escaped NIV-10 even at a concentration of 100 µg/mL, which provoked us to investigate the correlation between the hydrophobicity of RBD and antibody

binding.

Since XBB.1.5 has a new substitution at position 486, we made recombinant XBB RBD with a single amino acid substitution at a position 486 that can occur by a single nucleotide mutation from F486 and S486, and examined their antibody binding. XBB RBD with I486, L486, Y486, V486, A486, P486, T486, R486 mutation were examined for antibody binding by ECLIA. We also calculated the hydrophobicity of the mutant RBDs and confirmed that the SAP value and binding signal of each XBB RBD mutant to the antibodies were well correlated, except for the F486P mutant (XBB.1.5) to NIV-10 (newly added Fig. 7c). Structural modeling suggested the fitness of NIV-10 to XBB.1.5 was impeded by the proline mutation at position 486, mainly by a steric clash between P486 and the antibody (newly added Fig. S10a).

To address this limitation of NIV-10, we computationally designed a new antibody sequence based on NIV-10, emphasizing interaction energy, shape complementarity, and the number of hydrogen bonds at the antibody-antigen interface. This resulted in the creation of a NIV-10-based designed antibody, NIV-10/FD03. Recombinant NIV-10/FD03 antibody potently bound to XBB RBDs regardless of mutations at position 486, potently bound to other previous VOCs, and neutralized XBB and XBB.1.5 (newly added Fig. 7d, 7e, and 7f). These new data are now shown in Figure 7, Fig. S9, Fig. S10 and described on page 10 line 8 to page 11 line 13, page 12 line 27 to page 13 line 9.

In addition, to make the computationally designed antibody public, the sequence and theoretical model structure of the designed antibody in complex with XBB.1.5 RBD have been submitted to the Biological Structure Model Archive (BSM-Arc) under BSM-ID BSM00046 (<https://bsma.pdbj.org/entry/46>).

2. Another concern is that the escapability was evaluated using the ancestral virus to identify escapes, potentially overlooking the influence of epistatic effects if using an Omicron variant as the backbone.

We agree this point is a caveat in our study, and we have discussed this point as limitation of this study (page 11 lines 24-26).

3. - Title and abstract: The title and abstract of the study do not adequately emphasize the significance of the most important discovery, which is the distinctiveness of the NIV-10 antibody.

Thank you for this valuable comment. In light of additional neutralization data against recent Omicron

sublineages, we have changed the title and abstract in the revised manuscript to add the significance of computational design for broadly neutralizing antibodies.

4. - *Fig. 1a: consider revising the data in Panel A to reflect the evolution of the virus since April 2022.*

We have included updated data from Apr 2023 in Fig. 1a.

5. - *Fig. 1c: The legend states that 823 clones were analyzed, however, it appears that 947 antibodies were actually evaluated based on the Results section (line 5).*

We have examined 947 ancestral RBD binding clones, and among them, 823 clones bound to Beta RBD. We have modified the legend to clarify these 823 clones were double binders (page 27, lines 8-12).

6. - *Fig. 2b: For improved readability, it would be advisable to substitute the heatmap plot with a table or a graph that clearly presents the actual IC50 values.*

We have substituted the heatmap plot with a table in Fig. 2b.

7. - *Fig. 2c-f: The study lacks clear information on the dosing regimen, route, and timing of antibody administration in the in vivo challenge study, please make the necessary adjustments.*

We have added detailed information in the legend (page 28 lines 7-9, 11-12).

8. - *Fig. 3a: The effect of RBD mutations on the fitness of the virus is not clear in the figure presented. The statement that the NIV-10 escapes are solely caused by the mutation from G to P at site 485 is not entirely accurate, as other substitutions such as L455W/C or E484K also had an impact on binding, although to a lesser degree. It would be beneficial to determine which of these substitutions can occur as a result of a single nucleotide mutation.*

Thank you for this valuable comment. We have toned down the statement regarding G485P (page 6 lines 12-13), and changed the logo plot color by natural mutation frequencies (Fig. 3a). Also we have created a new supplementary table 4 which shows amino acid substitutions occur as a result of a single nucleotide mutation and described in page 6 line 15.

9. - *Fig 3b: The figure is difficult to read due to the small font size and legends, it is recommended to redo this figure for improved legibility.*

Thank you for the suggestion. We have enlarged the font size for Fig 3b.

10. - *Fig. 3c: the top five NIV-10 escapes tested (D405M, Y449P/K, L455C and G485P) are not all listed in Fig. 3a*

Figure 3a is presented based on epitope information of NIV-8, 10, -11 and -13 as described in the legend, and therefore D405 is not listed in Figure 3a. Top five NIV-10 escapes are presented in Figure 3b and Figure 3c.

11. - *Fig. 3d: as mentioned above the potential impact of epistatic effects, particularly with Omicron variants carrying mutations at position 486, on the effect of escape mutants and neutralization susceptibility should be taken into account. It may be beneficial to test these mutants on the Omicron backbone (against which the affinity of NIV-10 is reduced by more than 3-logs, as shown in Supplementary Table 3) or include a cautionary note in the interpretation of these results in the text.*

We appreciate this valuable comment. As we described in the reply for recent variants neutralization, we generated XBB RBD with mutations at position 486 which can occur as a single nucleotide mutation and examined their binding to NIV-10. Hydrophobicity of each RBD mutants and NIV-10 binding signal were nicely correlated except for the proline mutant. We show this new data in Fig. 7c, and described on page 10 lines 16-23.

12. - *Page 10, lines 27-28: The statement that "These data suggest that current mRNA vaccination cannot efficiently boost or maintain Y489S- B cells" may be considered too speculative. It may be appropriate to tone it down or remove it.*

Thank you for the comment. We removed the sentence.

13. - *Page 4, lines 1-2: Please provide an explanation for the choice of mutations and whether they represent the mutations with the greatest reduction in ACE2 binding.*

We chose mutations based on low frequency among the circulating variants and attenuated ACE2 binding ability (page 4 lines 8-11). We confirmed mutant RBDs with these mutations showed

undetectable binding affinities to human ACE2 in Supplementary fig. 1a (page 4 line 11-13).

14. - Page 4, line 2-3. *The sentence "Five amino acids were positioned in the RBS, suggesting that they are critical for ACE2 binding" appears to be tautological in nature.*

Thank you for the comment. We have removed the sentence.

15. - Page 4, line 14: *Please revise the text to accurately reflect the evolution of Omicron variants. BA.4/5 are no longer considered recent variants as they have been present for over six months and have been largely replaced by newer variants with different F486 mutations, such as XBB.1 (F486S), XBB.1.5 (F486P), BA.4.6 (F486S), and BA.2.75.2 (F486S).*

Thank you for the comment. We have updated the sentence (page 4 lines 22-23).

16. - Page 6, line 24: *please indicate that the mutation found in BA.4 and BA.5 is F486V*

We added the statement (page 7 line 2).

17. - Page 6, line 31: *remove "the" in "the neutralizing activity"*

We removed it.

18. - Page 6, line 6: *this is consistent with epistatic effects of other Omicron mutations, as shown in the paper by Bloom and colleagues (Starr et al, doi.org/10.1371/journal.ppat.1010951). Please consider adding this quote.*

Thank you for this suggestion. We believe this comment is related to page 7, line 6, and we have added the sentence and the reference accordingly (page 7 lines 17-18).

19. - Page 7, lines 7-8: *this statement is further supported by the recent data showing that the mutation of S486 in XBB.1 into P486 in XBB.1.5 resulted in a markedly increased affinity for huACE2, as reported in Yue et al. (10.1101/2023.01.03.522427).*

Thank you for this suggestion. We have added the sentence and the reference (page 7 lines 20-21).

20. - Page 8, line 8-9: *consider replacing "showed" with "recognized"*

We changed it accordingly (page 8 line 19).

21. - Page 10, line 10: The expression "the neck to the left shoulder" can be defined more clearly by replacing it with "the neck to the left shoulder RBS region."

Thank you for the suggestion. We changed it accordingly (page 12 line 1).

22. - Page 10, lines 15-17: The sentence regarding the intriguing outcomes obtained from MD simulations may need to be better articulated and expanded for clarity.

Thank you for the comment. We have added some descriptions regarding this point on page 9, lines 14-21.

23. - Page 10, line 2: this may be true for infections not just vaccinations.

Thank you for the suggestion. We believe this comment is related to page 11, line 2, and modified the sentence (page 12 lines 24-25).

24. - Fig. 3d: The plot is difficult to interpret due to indistinct colors, please consider improving it for better readability. The same for Figure 4 a-e plots.

Thank you for the suggestion. We have changed the colouring in Fig. 2c-f, 3d, 4a-e.

25. - Fig. 6d: consider highlighting NIV-10 mAb in the plot

We have highlighted the NIV-10 mAb data in red in Fig. 6d.

We greatly appreciate the helpful comments of the reviewer; they have improved the manuscript substantially. We sincerely hope that this revised revision meets the criteria for publication in the Nature Communications.

REVIEWERS' COMMENTS:

Reviewer #1 (Remarks to the Author):

Authors have adequately addressed all the comments, I'm happy to recommend this paper for publication.

Reviewer #2 (Remarks to the Author):

My comments have been sufficiently addressed. I would recommend publication of the revised manuscript.

Reviewer #3 (Remarks to the Author):

The authors have addressed my points. I am happy to recommend publication of the revised paper. Congratulations to the authors of this important work.

Point-by-point response

REVIEWERS' COMMENTS:

Reviewer #1 (Remarks to the Author):

Authors have adequately addressed all the comments, I'm happy to recommend this paper for publication.

Reviewer #2 (Remarks to the Author):

My comments have been sufficiently addressed. I would recommend publication of the revised manuscript.

Reviewer #3 (Remarks to the Author):

The authors have addressed my points. I am happy to recommend publication of the revised paper. Congratulations to the authors of this important work.

We again thank reviewers for their constructive comments on our manuscript.